# A cross-kingdom conserved ER-phagy receptor maintains endoplasmic reticulum homeostasis during stress

Madlen Stephani[1†], Lorenzo Picchianti[1,2†], Alexander Gajic[1], Rebecca Beveridge[2], Emilio Skarwan[1], Victor Sanchez de Medina Hernandez[1], Azadeh Mohseni[1], Marion Clavel[1], Yonglun Zeng[3], Christin Naumann[4], Mateusz Matuszkiewicz[1,5], Eleonora Turco[6], Christian Loefke[1], Baiying Li[3], Gerhard Dürnberger[1,2], Michael Schutzbier[1,2], Hsiao Tieh Chen[1,3], Alibek Abdrakhmanov[1], Adriana Savova[6], Khong-Sam Chia[1‡], Armin Djamei[1‡], Irene Schaffner[7], Steffen Abel[4], Liwen Jiang[3], Karl Mechtler[1,2], Fumiyo Ikeda[8,9], Sascha Martens[6], Tim Clausen[2,10]*, Yasin Dagdas[1]*

[1]Gregor Mendel Institute (GMI), Austrian Academy of Sciences, Vienna BioCenter (VBC), Vienna, Austria; [2]Research Institute of Molecular Pathology (IMP), Vienna BioCenter (VBC), Vienna, Austria; [3]School of Life Sciences, Centre for Cell and Developmental Biology and State Key Laboratory of Agrobiotechnology, The Chinese University of Hong Kong, School of Life Sciences, New Territories, Shatin, China; [4]Department of Molecular Signal Processing, Leibniz Institute of Plant Biochemistry, Halle, Germany; [5]Department of Plant Genetics, Breeding and Biotechnology, Warsaw University of Life Sciences-SGGW, Warsaw, Poland; [6]Department of Biochemistry and Cell Biology, Max Perutz Labs, University of Vienna, Vienna BioCenter (VBC), Vienna, Austria; [7]BOKU Core Facility Biomolecular & Cellular Analysis, University of Natural Resources and Life Sciences, Vienna, Austria; [8]Department of Molecular and Cellular Biology, Medical Institute of Bioregulation, Kyushu University, Fukuoka, Japan; [9]Institute of Molecular Biotechnology of the Austrian Academy of Sciences (IMBA), Vienna BioCenter (VBC), Vienna, Austria; [10]Medical University of Vienna, Vienna, Austria

*For correspondence:
tim.clausen@imp.ac.at (TC);
yasin.dagdas@gmi.oeaw.ac.at (YD)

†These authors contributed equally to this work

Present address: ‡Department of Breeding Research, Leibniz Institute of Plant Genetics and Crop Plant Research (IPK), Gatersleben, Germany

**Abstract** Eukaryotes have evolved various quality control mechanisms to promote proteostasis in the endoplasmic reticulum (ER). Selective removal of certain ER domains via autophagy (termed as ER-phagy) has emerged as a major quality control mechanism. However, the degree to which ER-phagy is employed by other branches of ER-quality control remains largely elusive. Here, we identify a cytosolic protein, C53, that is specifically recruited to autophagosomes during ER-stress, in both plant and mammalian cells. C53 interacts with ATG8 via a distinct binding epitope, featuring a shuffled ATG8 interacting motif (sAIM). C53 senses proteotoxic stress in the ER lumen by forming a tripartite receptor complex with the ER-associated ufmylation ligase UFL1 and its membrane adaptor DDRGK1. The C53/UFL1/DDRGK1 receptor complex is activated by stalled ribosomes and induces the degradation of internal or passenger proteins in the ER. Consistently, the C53 receptor complex and ufmylation mutants are highly susceptible to ER stress. Thus, C53 forms an ancient quality control pathway that bridges selective autophagy with ribosome-associated quality control in the ER.

**eLife digest** For cells to survive they need to be able to remove faulty or damaged components. The ability to recycle faulty parts is so crucial that some of the molecular machinery responsible is the same across the plant and animal kingdoms. One of the major recycling pathways cells use is autophagy, which labels damaged proteins with molecular tags that say 'eat-me'. Proteins called receptors then recognize these tags and move the faulty component into vesicles that transport the cargo to a specialized compartment that recycles broken parts.

Cells make and fold around 40% of their proteins at a site called the endoplasmic reticulum, or ER for short. However, the process of folding and synthesizing proteins is prone to errors. For example, when a cell is under stress this can cause a 'stall' in production, creating a build-up of faulty, partially constructed proteins that are toxic to the cell. There are several quality control systems which help recognize and correct these errors in production. Yet, it remained unclear how autophagy and these quality control mechanisms are linked together.

Here, Stephani, Picchianti et al. screened for receptors that regulate the recycling of faulty proteins by binding to the 'eat-me' tags. This led to the identification of a protein called C53, which is found in both plant and animal cells. Microscopy and protein-protein interaction tests showed that C53 moves into transport vesicles when the ER is under stress and faulty proteins start to build-up. Once there, C53 interacts with two proteins embedded in the wall of the endoplasmic reticulum. These proteins form part of the quality control system that senses stalled protein production, labelling the stuck proteins with 'eat-me' tags. Together with C53, they identify and remove half-finished proteins before they can harm the cell.

The fact that C53 works in the same way in both plant and human cells suggests that many species might use this receptor to recycle stalled proteins. This has implications for a wide range of research areas, from agriculture to human health. A better understanding of C53 could be beneficial for developing stress-resilient crops. It could also aid research into human diseases, such as cancer and viral infections, that have been linked to C53 and its associated proteins.

## Introduction

Autophagy is an intracellular degradation process where eukaryotic cells remove harmful or unwanted cytoplasmic contents to maintain cellular homeostasis (*Dikic and Elazar, 2018*; *Klionsky and Deretic, 2011*; *Marshall and Vierstra, 2018*). Recent studies have shown that autophagy is highly selective (*Johansen and Lamark, 2020*; *Stolz et al., 2014*) and is mediated by receptors that recruit specific cargo, such as damaged organelles or protein aggregates. Autophagy receptors and their cargo are incorporated into the growing phagophore through interaction with ATG8, a ubiquitin-like protein that is conjugated to the phagophore upon activation of autophagy (*Stolz et al., 2014*; *Zaffagnini and Martens, 2016*). The phagophore grows and eventually forms a double-membrane vesicle termed the autophagosome. Autophagosomes then carry the autophagic cargo to lytic compartments for degradation and recycling. Selective autophagy receptors interact with ATG8 via conserved motifs called the ATG8 interacting motif (AIM) or LC3-interacting region (LIR) (*Birgisdottir et al., 2013*). In contrast to mammals and yeast, cargo receptors that mediate organelle recycling remains mostly elusive in plants (*Stephani and Dagdas, 2020*).

The endoplasmic reticulum (ER) is a highly dynamic heterogeneous cellular network that mediates folding and maturation of ~40% of the proteome (*Walter and Ron, 2011*; *Sun and Brodsky, 2019*). Proteins that pass through the ER include all secreted and plasma membrane proteins and majority of the organellar proteins. This implies, ER could handle up to a million client proteins in a cell every minute (*Karagöz et al., 2019*). Unfortunately, the folding process is inherently error prone and misfolded proteins are toxic to the cell (*Sun and Brodsky, 2019*; *Karagöz et al., 2019*; *Fregno and Molinari, 2019*). To maintain the proteostasis in the ER, eukaryotes have evolved dedicated quality control mechanisms that closely monitor, and if necessary, trigger the removal of terminally misfolded proteins. Degradation of the faulty proteins is mediated by proteasomal and vacuolar degradation pathways (*Fregno and Molinari, 2019*).

One of the main vacuolar/lysosomal degradation processes is ER-phagy. It has emerged as a major quality control pathway, and defects in ER-phagy is linked to various diseases (*Chino and*

*Mizushima, 2020*; *Hübner and Dikic, 2020*; *Stolz and Grumati, 2019*; *Wilkinson, 2020*). ER-phagy involves cargo receptors that mediate removal of certain regions of the ER via autophagy. Several ER-resident ER-phagy receptors have been identified. These include Fam134B, RTN3L, ATL3, Sec62, CCPG1, and TEX264 in mammals and ATG39 and ATG40 in yeast (*Khaminets et al., 2015*; *Grumati et al., 2017*; *Chen et al., 2019*; *Fumagalli et al., 2016*; *Smith et al., 2018*; *An et al., 2019*; *Chino et al., 2019*; *Mochida et al., 2015*). A recent study showed reticulon proteins could also function as ER-phagy receptors in plants (*Zhang et al., 2020*). In addition, CALCOCO1 and Epr1 have been recently identified as cytosolic ER-phagy receptors that associate with ER-resident VAP proteins to recycle ER tubules (*Nthiga et al., 2020*; *Zhao et al., 2020*). Altogether, these receptors are activated during starvation or stress conditions and work together to remodel the highly heterogeneous and dynamic ER network to maintain proteostasis. Despite the emerging links, how ER-phagy cross-talks with the core ER quality control pathways remains largely unknown (*Chino and Mizushima, 2020*; *Dikic, 2018*).

Here, using a peptide-competition coupled affinity proteomics screen, we identified a highly conserved cytosolic protein, C53, that is specifically recruited into autophagosomes during ER stress. C53 interacts with plant and mammalian ATG8 isoforms via a non-canonical ATG8 interacting motif (AIM), termed shuffled AIM (sAIM). C53 is recruited to the ER by forming a ternary receptor complex with the UFL1, the E3 ligase that mediates ufmylation, and its ER membrane adaptor DDRGK1 (*Gerakis et al., 2019*). C53-mediated autophagy is activated upon ribosome stalling during co-translational protein translocation and results in the degradation of specific ER proteins.

## Results

### C53 interacts with plant and mammalian ATG8 isoforms in an ER-stress dependent manner

To identify specific cargo receptors that mediate selective removal of ER compartments during proteotoxic stress, we performed an immunoprecipitation coupled to mass spectrometry (IP-MS) screen to identify AIM-dependent ATG8 interactions triggered by ER stress. We hypothesized that a synthetic AIM peptide that has higher affinity for ATG8 can outcompete, and thus reveal, AIM-dependent ATG8 interactors. To identify this synthetic peptide, we performed a peptide array analysis that revealed the AIM *wt* peptide (*Figure 1—figure supplement 1A,B*; *Supplementary file 1*). Using isothermal titration calorimetry (ITC), we showed that the AIM *wt* binds ATG8 with nanomolar affinity ($K_D = \sim 700$ nM), in contrast to the AIM mutant peptide (AIM *mut*), which does not show any binding (*Figure 1—figure supplement 1C–D*) or the low micromolar-range affinities measured for most cargo receptors (*Zaffagnini and Martens, 2016*). As plants have an expanded set of ATG8 proteins, we first tested if any of the ATG8 isoforms specifically responded to ER stress induced by tunicamycin (*Kellner et al., 2017*). Tunicamycin inhibits glycosylation and leads to proteotoxic stress at the ER (*Bernales et al., 2006*). Quantification of ATG8 puncta in transgenic seedlings expressing GFP-ATG8A-I revealed that tunicamycin treatment significantly induced all nine ATG8 isoforms (*Figure 1—figure supplement 2*). Since all ATG8 isoforms were induced and ATG8E has a broad expression pattern, we chose ATG8E, and performed peptide competition coupled IP-MS analysis (See Materials and methods for detailed description). In addition to well-known AIM dependent ATG8 interactors such as ATG4 (Autophagy related gene 4) and NBR1 (Neighbour of BRCA1) (*Wild et al., 2014*), our analyses revealed that the highly conserved cytosolic protein C53 (aliases: CDK5RAP3, LZAP, IC53, HSF-27) is an AIM-dependent ATG8 interactor (*Figure 1A*, *Supplementary file 2*, *Figure 1—figure supplement 3*).

To confirm our IP-MS results, we performed in vitro pull-down experiments. *Arabidopsis thaliana* (At) C53 specifically interacted with GST-ATG8A, and this interaction was outcompeted with the AIM *wt*, but not AIM *mut* peptide. Consistently, ATG8 receptor accommodating site mutations (LDS − LIR Docking Site) prevented C53 binding (*Figure 1B*). We extended our analysis to all Arabidopsis ATG8 isoforms and showed that AtC53 interacts with eight of nine Arabidopsis isoforms (*Figure 1C*). To probe for evolutionary conservation of C53-ATG8 interaction, we tested the orthologous proteins from the basal land plant *Marchantia polymorpha* (Mp) and showed that MpC53 interacts with one of two Marchantia ATG8 isoforms (*Figure 1—figure supplement 4*). As C53 is highly conserved in multicellular eukaryotes and has not been characterized as an ATG8 interactor in

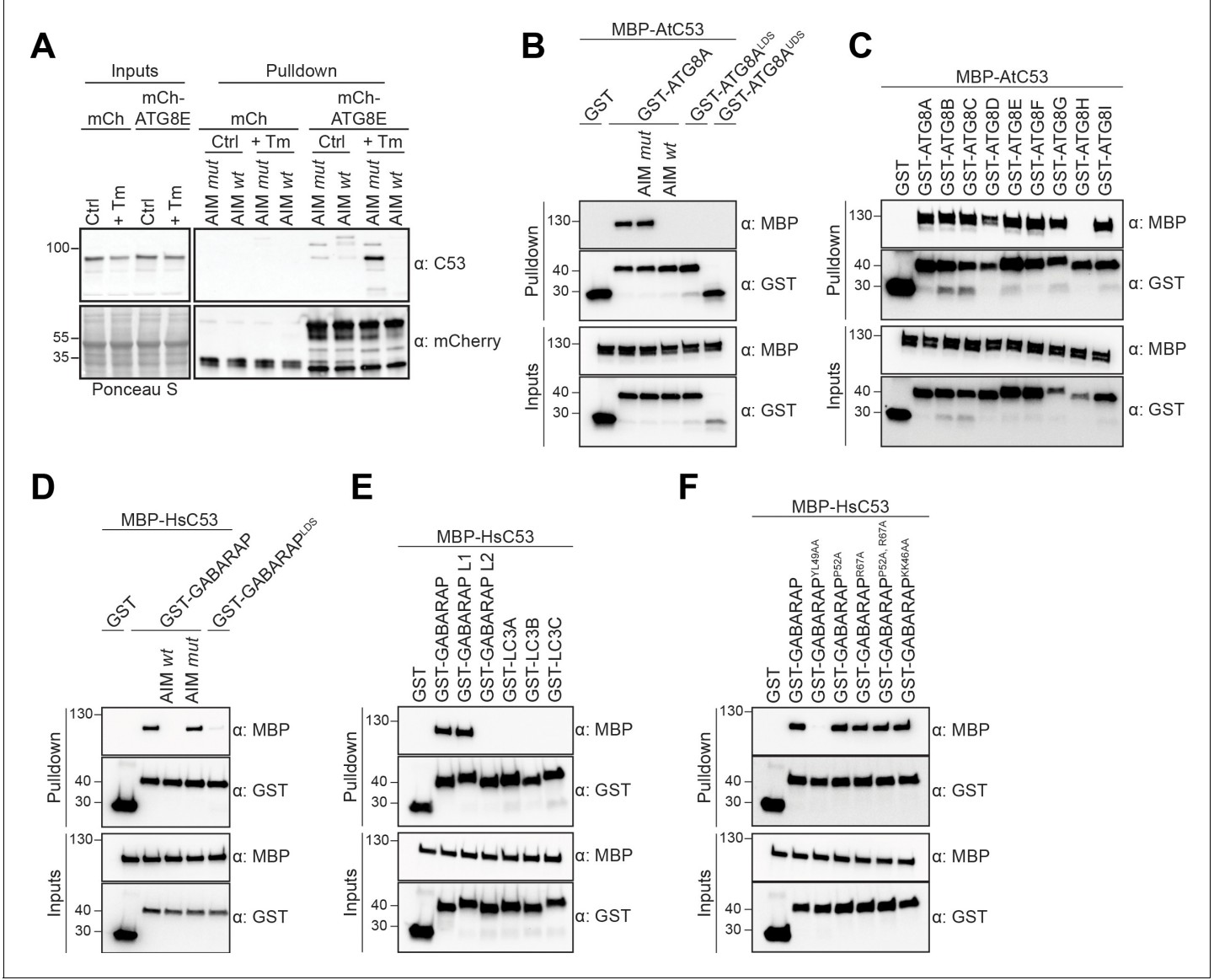

**Figure 1.** C53 binds ATG8 in an AIM dependent manner via the LIR Docking Site (LDS). (**A**) Peptide competition coupled in vivo pull-down revealed C53 as an AIM dependent ATG8 interactor. In vivo co-immunoprecipitation of extracts of Arabidopsis seedlings expressing mCherry alone or mCherry-ATG8E incubated in Control or 10 µg/ml tunicamycin (Tm) containing media. The peptides AIM *wt* and AIM *mut* were added to a final concentration of 200 µM. Input and bound proteins were visualized by immunoblotting with anti-mCherry and anti-C53 antibodies. (**B**) AtC53 interact with ATG8A in an AIM-dependent manner. Bacterial lysates containing recombinant protein were mixed and pulled down with glutathione magnetic agarose beads. The peptides AIM *wt* and AIM *mut* were added to a final concentration of 200 µM. (**C**) AtC53 interacts with AtATG8 in an isoform specific manner. In vitro pull down with all ATG8 isoforms of *Arabidopsis thaliana* (At) shows that AtC53 can interact with eight out of nine ATG8 isoforms. (**D**) HsC53 interact with GABARAP in an AIM-dependent manner. Bacterial lysates containing recombinant protein were mixed and pulled down with glutathione magnetic agarose beads. The peptides AIM *wt* and AIM *mut* were added to a final concentration of 200 µM. (**E**) HsC53 interacts with GABARAP and GABARAP L1. Bacterial lysates containing recombinant protein were mixed and pulled down with glutathione magnetic agarose beads. (**F**) HsC53 interacts with GABARAP via the LIR Docking Site (LDS). Mutating the W site to a YL49AA mutation (LDS) (*Marshall et al., 2019*) prevents binding of GABARAP to C53. However, mutating the L position to P52A or R67A (*Marshall et al., 2019*), or mutating KK64AA (which mediates the interaction with the atypical LIR motif found in UBA5 [*Huber et al., 2019*]) did not prevent C53 binding. Bacterial lysates containing recombinant protein were mixed and pulled down with glutathione magnetic agarose beads. Input and bound proteins were visualized by immunoblotting with anti-GST and anti-MBP antibodies. LDS = LIR Docking-Site mutant (*Marshall et al., 2019*; UDS = Ubiquitin Docking Site mutant *Marshall et al., 2019*).

The online version of this article includes the following figure supplement(s) for figure 1:

**Figure supplement 1.** Identification of high affinity AIM peptides for peptide competition coupled immunoprecipitation mass spectrometry and in vitro pull-down experiments.

**Figure supplement 2.** All Arabidopsis ATG8 isoforms are induced by tunicamycin-triggered ER stress.

*Figure 1 continued on next page*

*Figure 1 continued*

**Figure supplement 3.** Unrooted maximum likelihood phylogenetic tree of C53 homologs.
**Figure supplement 4.** MpC53 interacts with MpATG8 isoforms in a specific manner.

mammals, we tested whether human C53 (HsC53) interacts with human ATG8 isoforms (LC3A-C, GABARAP, -L1, -L2). HsC53 interacted with GABARAP and GABARAPL1 in an AIM-dependent manner via the LIR docking site, similar to plant C53 homologs (*Figure 1D,E*). Of note, we have also tested other modes of ATG8 binding such as the recently identified UDS or the hydrophobic pocket accommodating the atypical LIR motif found in ufmylation enzyme UBA5 (*Marshall et al., 2019*; *Huber et al., 2019*). The UDS mutation rendered ATG8A unstable (*Figure 1B*), whereas mutating the atypical LIR accommodating site did not affect C53 binding (*Figure 1F*). Altogether, these data suggest that C53-ATG8 interaction is conserved across kingdoms and mediated via the LIR Docking Site.

In order to examine the in vivo link between C53 and ATG8 function, we generated transgenic AtC53-mCherry Arabidopsis lines and measured autophagic flux during ER stress. Without stress, AtC53 displayed a diffuse pattern in the cell, partially overlapping with the ER marker GFP-HDEL (*Figure 2—figure supplement 1A*). Similarly, upon carbon starvation (-C, *Figure 2A*), which is commonly used to activate bulk autophagy, AtC53-mCherry remained mostly diffuse (*Marshall and Vierstra, 2018*). However, tunicamycin (Tm) treatment led to a rapid increase in AtC53 puncta as observed in both native promoter driven and ubiquitin 10 promoter driven transgenic lines. The C53 puncta did not colocalize with HDEL-GFP puncta formed during ER stress, suggesting C53 puncta are highly specific (*Figure 2—figure supplement 1A,B*). The number of puncta was further increased upon concanamycin A (ConA) treatment that inhibits vacuolar degradation, suggesting that AtC53 puncta are destined for vacuoles (*Figure 2A*). The AtC53 puncta disappeared when AtC53-mCherry lines were crossed into core autophagy mutants *atg5* and *atg2*, confirming that formation of AtC53 puncta is dependent on macroautophagy (*Figure 2A*). Consistent with this, other ER-stressors such as phosphate starvation, cyclopiazonic acid (CPA), and dithiothreitol (DTT) treatments also induced AtC53 puncta (*Figure 2—figure supplement 1C*; *Fumagalli et al., 2016*; *Smith et al., 2018*; *Naumann et al., 2019*). The AtC53 puncta co-localized with GFP-ATG8A and GFP-ATG11, indicating that they are autophagosomes (*Figure 2B*, *Figure 2—figure supplement 2A*). Moreover, as recently shown for other selective autophagy receptors, AtC53 and HsC53 directly interacted with the mammalian ATG11 homolog FIP200 (PTK2/FAK family-interacting protein of 200 kDa) (*Figure 2—figure supplement 2B*; *Lahiri and Klionsky, 2018*; *Turco et al., 2019*; *Ravenhill et al., 2019*; *Vargas et al., 2019*). Ultrastructural analysis using immunogold labelling showed that C53 is associated with ER under non-stress conditions, consistent with previous findings showing C53 associates with ER membrane proteins (*Yang et al., 2019*). Electron micrographs also showed that AtC53 is recruited to autophagosomes during ER stress, consistent with our live cell imaging results (*Figure 2C*, *Figure 2—figure supplement 3*). Similar to plant proteins, transfected HsC53-GFP co-localized with mCherry-GABARAP upon tunicamycin treatment in HeLa cells. The number of HsC53 puncta increased upon bafilomycin (BAF) treatment, which inhibits lysosomal degradation; suggesting that HsC53 puncta eventually fuse with lysosomes (*Figure 2D*). To support our imaging-based autophagic flux assays, we also performed western blot based autophagic flux analyses, using antibodies raised against AtC53 and HsC53. These autophagic flux assays further demonstrated ER-stress-specific autophagic degradation of AtC53 and HsC53 (*Figure 3*).

## C53-ATG8 interaction is mediated by non-canonical shuffled ATG8 interacting motifs (sAIM)

Having validated C53 as an autophagy substrate, we next sought to identify its ATG8-interacting motif (AIM). For this purpose, we reconstituted the binary complex in vitro and determined the stoichiometry of the C53-ATG8 interaction by native mass spectrometry (nMS). Both HsC53 and AtC53 formed 1:1 and 1:2 complexes with GABARAP and ATG8A, respectively; pointing to the existence of multiple binding epitopes (*Figure 4A*). Initially, we tested all predicted canonical AIMs in AtC53. However, even the pentuple AIM mutant bound at similar levels to ATG8, suggesting non-canonical AIMs mediate the C53-ATG8 interaction (*Figure 4—figure supplement 1*). To narrow down the

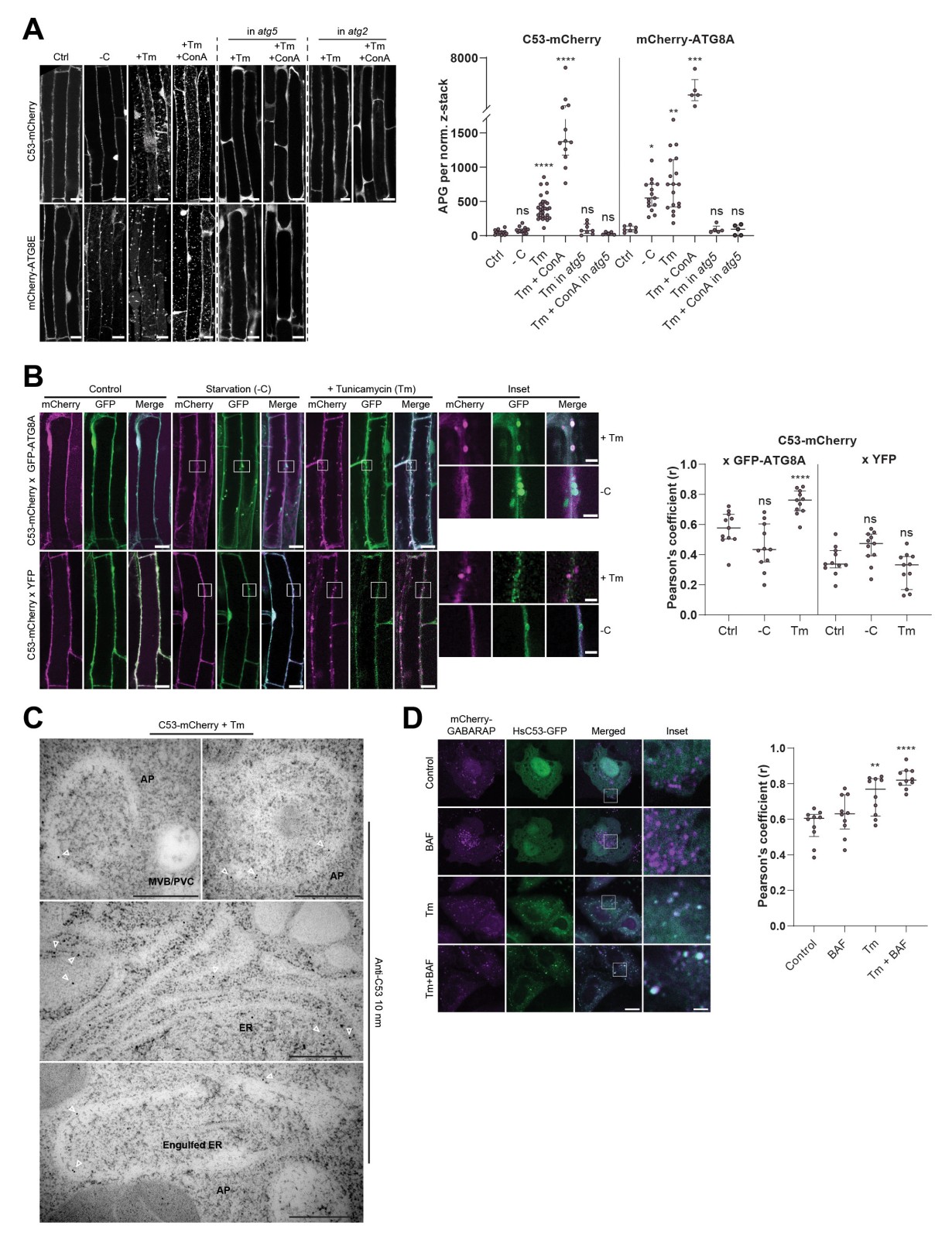

**Figure 2.** C53 is recruited to autophagosomes during ER-stress and undergoes ER-stress specific autophagic degradation. (**A**) AtC53 is specifically recruited to puncta upon ER stress and this depends on ATG5 and ATG2. *Left Panel*, representative confocal images of transgenic Arabidopsis seedlings expressing C53-mCherry and mCherry-ATG8E in Col-0 wild type, *atg5* and *atg2* mutant backgrounds. Six-day-old seedlings were incubated in either control, sucrose (-C)-deficient, tunicamycin (10 μg/ml), or tunicamycin (Tm, 10 μg/ml) + Concanamycin (ConA, 1 μM) containing media. Scale

*Figure 2 continued on next page*

*Figure 2 continued*

bars, 10 μm. *Right Panel*, Quantification of the autophagosomes (APG) per normalized Z-stacks. Bars represent the mean (± SD) of at least 10 biological replicates. (B) AtC53 puncta colocalize with GFP-ATG8A-labeled autophagosomes during ER stress. *Left Panel*, Co-localization analyses of single plane confocal images obtained from transgenic Arabidopsis roots co-expressing C53-mCherry (magenta) with GFP-ATG8A or YFP alone (green). Four-day-old seedlings were incubated in either control, sucrose deficient (-C), or tunicamycin containing media. Scale bars, 20 μm. Inset scale bars, 2 μm. *Right Panel*, Pearson's Coefficient (r) analysis of the colocalization of C53-mCherry with GFP-ATG8A or YFP alone. Bars represent the mean (± SD) of at least five biological replicates. (C) Electron micrographs showing that C53 localizes to the ER and autophagosomes during ER stress. Immunogold labeling of high-pressure frozen, 5-day-old *Arabidopsis* roots treated with 300 ng/ml tunicamycin for 6 hr. Arrowheads indicate 10 nm gold particles. Scale bars, 500 nm. ER = Endoplasmic reticulum, AP = autophagosome, MVB/PVC = multivesicular body. (D) HsC53 puncta colocalize with mCherry-GABARAP labelled autophagosomes during ER stress. *Left Panel*, Confocal images of PFA fixed HeLa cells transiently expressing C53-GFP (green) and mCherry-GABARAP (magenta). Cells were either untreated (Control) or treated with tunicamycin (Tm) or Tm + Bafilomycin (BAF). Scale bar, 20 μm. Inset scale bar, 2 μm. *Right Panel*, Pearson's Coefficient analysis of the colocalization of HsC53-GFP with mCherry-GABARAP under control and Tm-treated conditions. Bars represent the mean (± SD) of at least five biological replicates. Significant differences are indicated with * when p value ≤ 0.05, ** when p value ≤ 0.01, and *** when p value ≤ 0.001.

The online version of this article includes the following figure supplement(s) for figure 2:

**Figure supplement 1.** Analysis of AtC53 puncta under various stress conditions revealed induction of C53 puncta upon ER stress.

**Figure supplement 2.** C53 binds selective autophagy adaptor ATG11.

**Figure supplement 3.** Electron micrographs showing that C53 localizes to the ER and autophagosomes during ER stress.

ATG8-binding region of C53, we performed in vitro pull downs using truncated proteins. C53 contains an intrinsically disordered region (IDR) that bridges two α-helical domains located at the N and C termini. In vitro pull downs revealed that the IDR is necessary and sufficient to mediate ATG8 binding, as also confirmed with ITC and nMS experiments (*Figure 4B–D*, *Figure 4—figure supplement 2*). Multiple sequence alignment of the C53-IDR uncovered three highly conserved sites with the consensus sequence 'IDWG', representing a shuffled version of the canonical AIM sequence (W/F/Y-X-X-L/I/V) (*Figure 4C*, *Figure 4—figure supplement 3*). Mutational analysis of the three shuffled AIM sites in HsC53 and AtC53 revealed the importance of the sAIM epitopes for binding to GABARAP and ATG8, respectively; though in AtC53, an additional canonical AIM had to be mutated to fully abrogate the binding (*Figure 5A*). ITC experiments with the purified IDRs, as well as nMS and surface plasmon resonance (SPR) experiments with full-length proteins, also supported sAIM-mediated ATG8-binding for both HsC53 and AtC53 (*Figure 5B,C*, *Figure 5—figure supplement 1*). Circular dichroism spectroscopy showed that sAIM mutants had very similar secondary structures to the wild-type proteins, suggesting that lack of ATG8 binding is not due to misfolding (*Figure 5—figure supplement 1C*). To verify our in vitro results in vivo, we analyzed the subcellular distribution of sAIM mutants in transgenic Arabidopsis lines and transfected HeLa cells. Confocal microscopy analyses showed that C53$^{sAIM}$ mutants were not recruited into autophagosomes and had diffuse localization patterns upon ER stress induction (*Figure 5B,C*). Altogether these biochemical and cell biological analyses show that C53 is recruited to the autophagosomes by interacting with ATG8 via the non-canonical sAIMs.

## C53 is activated by ribosome stalling during co-translational protein translocation

Next, we looked for client proteins subject to C53-mediated autophagy. Quantitative proteomics analyses of wild type and AtC53 mutant lines revealed that AtC53 mediates degradation of ER resident proteins as well as proteins passing the ER to the cell wall, apoplast, and lipid droplets (*Figure 6*, *Supplementary file 3*, *4*). These data are consistent with a recent study, showing that ER-resident proteins accumulate in a conditional mutant of mouse C53 (*Yang et al., 2019*). Since C53 is a cytosolic protein, we then explored how it senses proteotoxic stress in the ER lumen, considering four likely scenarios: C53 may collaborate with (*i*) a sensor of the unfolded protein response (UPR) (*Karagöz et al., 2019*) or (*ii*) a component of the ER-associated degradation pathway (ERAD) (*Sun and Brodsky, 2019*). Alternatively, it may sense clogged translocons caused by (*iii*) ribosome stalling triggered during co-translational protein translocation (*Wang et al., 2020*) or (*iv*) aberrant signal recognition particle (SRP) independent post-translational protein translocation events (*Ast et al., 2016*; *Figure 7A*). In plants, there are two major UPR branches: the Ire1 pathway and bZIP17/28 pathway (*Pastor-Cantizano et al., 2020*). To test the connection with the UPR system, we

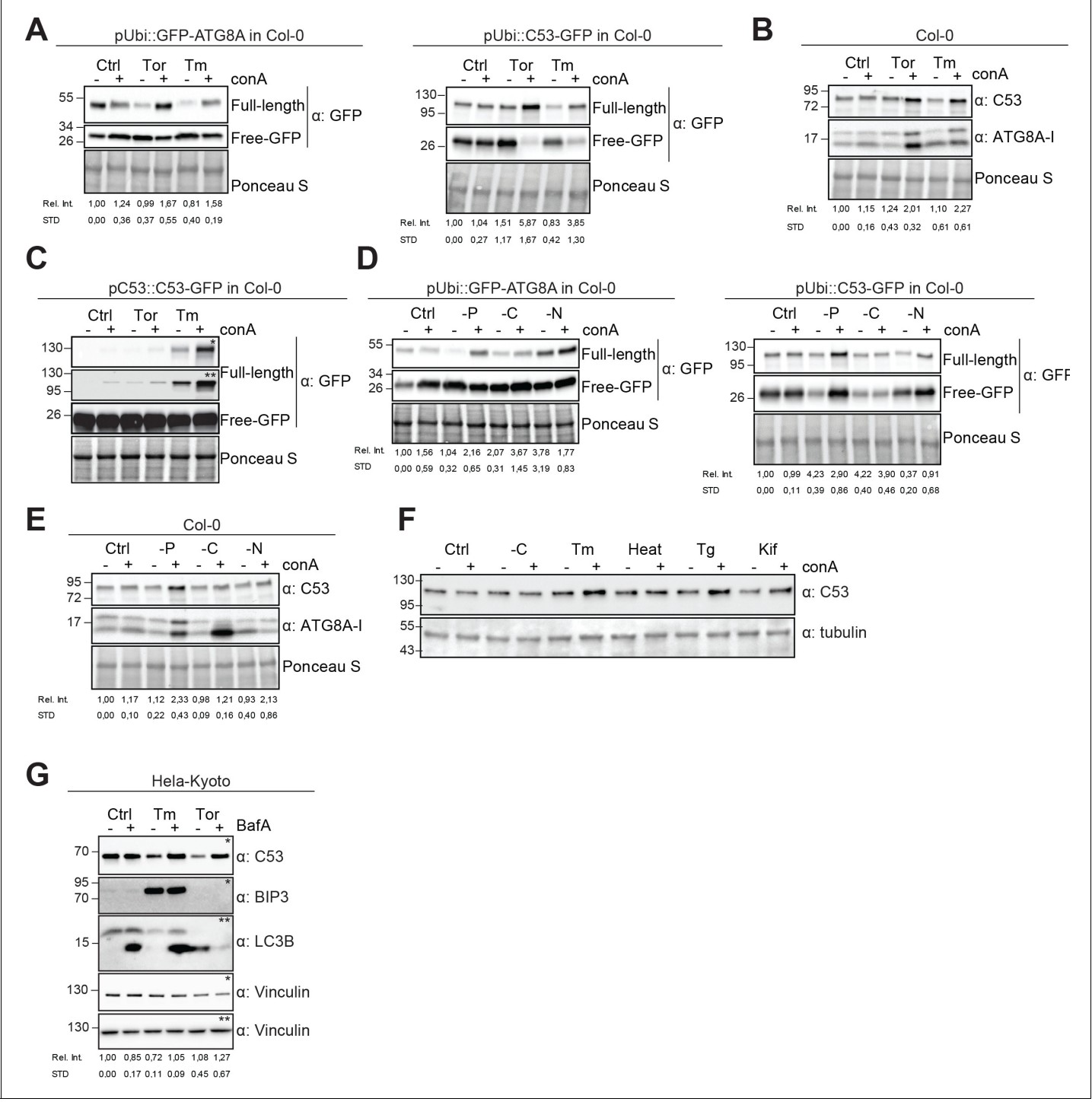

**Figure 3.** Autophagic flux analysis of AtC53 and HsC53 show that C53 autophagic flux is induced during ER stress. (**A–C**) AtC53 flux is induced by Torin and tunicamycin treatment. (**A**) Autophagic flux analysis of transgenic pUbi::AtC53-GFP (right panel) and pUbi::GFP-ATG8A (left panel) seedlings. (**B**) Autophagic flux analysis of endogenous AtC53 and ATG8, using AtC53 and ATG8 antibodies, respectively. (**C**) Autophagic flux analysis of transgenic pAtC53::AtC53-GFP seedlings. Col-0 or transgenic seedlings were incubated in control media or media containing 9 μM Torin1 (Tor) or 10 μg/ml tunicamycin (Tm). In addition, each treatment was supplied with 1 μm concanamycin A (conA) to visualize vacuolar degradation. Representative Western blots are displayed. Full-length and free GFP-bands from the same blot were separated due to different exposure times. In (**C**), * and ** correspond to short and long exposures of the same blot, respectively. Quantification of the relative intensities (Rel. Int.) of the protein bands were normalized for the total protein level of the lysate (Ponceau S). Average C53 levels and SD for n = 3 are shown. (**D-E**) AtC53 flux is specifically induced upon phosphate starvation. (**D**) Autophagic flux analysis of transgenic pUbi::AtC53-GFP (right panel) and pUbi::GFP-ATG8A (left panel) seedlings under carbon, nitrogen,

*Figure 3 continued on next page*

Figure 3 continued

and phosphate starvation conditions. (**E**) Autophagic flux analysis of endogenous AtC53 and ATG8, using AtC53 and ATG8 antibodies, respectively. Col-0 or transgenic seedlings were incubated in control media or media depleted with sucrose (-C), nitrogen (-N) or phosphate (-P). In addition, each treatment was supplied with 1 μm concanamycin A (conA) to visualize vacuolar degradation. Representative western blots are displayed. Full-length and free GFP-bands from the same blot were separated due to different exposure times. Quantification of the relative intensities (Rel. Int.) of the protein bands were normalized for the total protein level of the lysate (Ponceau S). Average C53 levels and SD for n = 3 are shown. (**F**) AtC53 autophagic flux is induced by various ER stress inducing conditions. Western blot analysis of Arabidopsis transgenic seedlings expressing AtC53-GFP incubated in either control (Ctrl), sucrose -deficient medium (-C), 10 μg/ml tunicamycin (Tm), 3 hr at 37°C (Heat), 2.5 μM Thapsigargin (Tg), or 50 μM Kifunensine (Kif). In addition, each treatment was supplied with 1 μm concanamycin A (conA) to visualize vacuolar degradation. (**G**) HsC53 flux is induced by Torin and tunicamycin treatment. Western blot analysis of HeLa whole cell lysates. Cells were either left untreated or treated for 16 hr with 2.5 μg/ml tunicamycin (Tm) or 1.5 μM Torin (Tor) and subsequently given a recovery period of 2 hr in presence or absence of 100 nM Bafilomycin A1 (BAF). C53 and BIP3 blots were run on 4–20% gradient gels and transferred to nitrocellulose membranes, LC3B blots were run on 15% gels and transferred to PVDF membranes. (* or ** indicate corresponding membranes). Quantification of the relative intensities (Rel. Int.) of the protein bands were normalized for the total protein level of the lysate (Vinculin). Average C53 levels and SD for n = 3 are shown.

performed autophagic flux assays. AtC53 flux was already higher than wild type in Arabidopsis UPR sensor mutants *ire1a/b* and *bzip17/28*, consistent with elevated ER stress levels in these mutants (*Figure 7B,C*; *Koizumi et al., 2001*; *Kim et al., 2018*). Furthermore, C53 puncta induced by tunicamycin treatment did not colocalize with Ire1b-YFP oligomers (*Figure 7D*). Finally, inhibition of Ire1 activity in HeLa cells using chemical inhibitors 4μ8c or KIRA6 also increased HsC53 puncta (*Figure 7E*). Together these data indicate that recruitment of C53 to the autophagosomes does not depend on UPR sensors (*Maly and Papa, 2014*). Next, we performed colocalization analyses using model ERAD substrates. In transgenic plant lines expressing model ERAD substrates, the client proteins did not colocalize with AtC53 puncta (*Figure 7—figure supplement 1A*; *Shin et al., 2018*). Likewise, the model mammalian ERAD substrates GFP-CFTRΔF508 (ERAD-C), A1AT^NHK-GFP (ERAD-L), and INSIG1-GFP (ERAD-M) only partially colocalized with HsC53 puncta in HeLa cells (*Figure 7—figure supplement 1B*), suggesting C53-mediated autophagy may cross-talk with the ERAD pathway (*Leto et al., 2019*).

Next, we tested the effect of clogged translocons on C53 function. Remarkably, HsC53 significantly colocalized with the ER-targeted poly-lysine construct ER-K20 that leads to ribosome stalling (*Wang et al., 2020*), but not with an SRP-independent translocon clogger (*Ast et al., 2016*), despite both leading to a blockage at the Sec61 translocon (*Figure 7—figure supplement 2A*). To further corroborate these findings, we tested a suite of translation inhibitors that block different steps in translation. Consistent with C53 responding to ribosome stalling, elongation inhibitors such as Anisomycin, Emetine or Puromycin induced AtC53 puncta, whereas initiation inhibitors Harringtonine or Hygromycin B did not have any effect. All inhibitors triggered mCherry-ATG8A puncta formation, suggesting the effect caused by elongation inhibitors is specific to C53 (*Figure 7—figure supplement 2B*). HsC53 puncta were also induced by anisomycin treatment (*Figure 7—figure supplement 2C*). Consistently, silencing of HsC53 using shRNA significantly reduced lysosomal trafficking of ER-K20 (*Figure 7—figure supplement 2D*; *Wamsley et al., 2017*). These data suggest that C53 is activated upon ribosome stalling during co-translational protein translocation and mediates autophagic degradation of the stalled nascent chain.

## C53 forms a heteromeric receptor complex with the ufmylation E3 ligase UFL1 and its membrane adaptor DDRGK1

How is C53 recruited to the ER during ribosome stalling? Notably, C53 has been previously linked to UFL1, an E3 ligase that mediates ufmylation of stalled, ER-bound ribosomes, modifying ribosomal protein RPL26 (*Wang et al., 2020*; *Walczak et al., 2019*). To test if C53 is a part of a higher order receptor complex, we analysed the interaction of C53 with UFL1 and its ER membrane adaptor DDRGK1 (*Gerakis et al., 2019*). We were able to observe both DDRGK1 and HsC53 in a single UFL1 pull down experiment (*Figure 8—figure supplement 1A*). Further in vitro pull-down assays and yeast two hybrid analyses with the plant proteins showed that AtUFL1 directly interacts with AtC53 and AtDDRGK1 (*Figure 8A*, *Figure 8—figure supplement 1B*). Consistently, AtC53 associates with DDRGK1 and UFL1 in in vivo coimmunoprecipitations and affinity purification mass spectrometry experiments (*Figure 8B*, *Supplementary file 5*). Furthermore, co-localization of UFL1 and

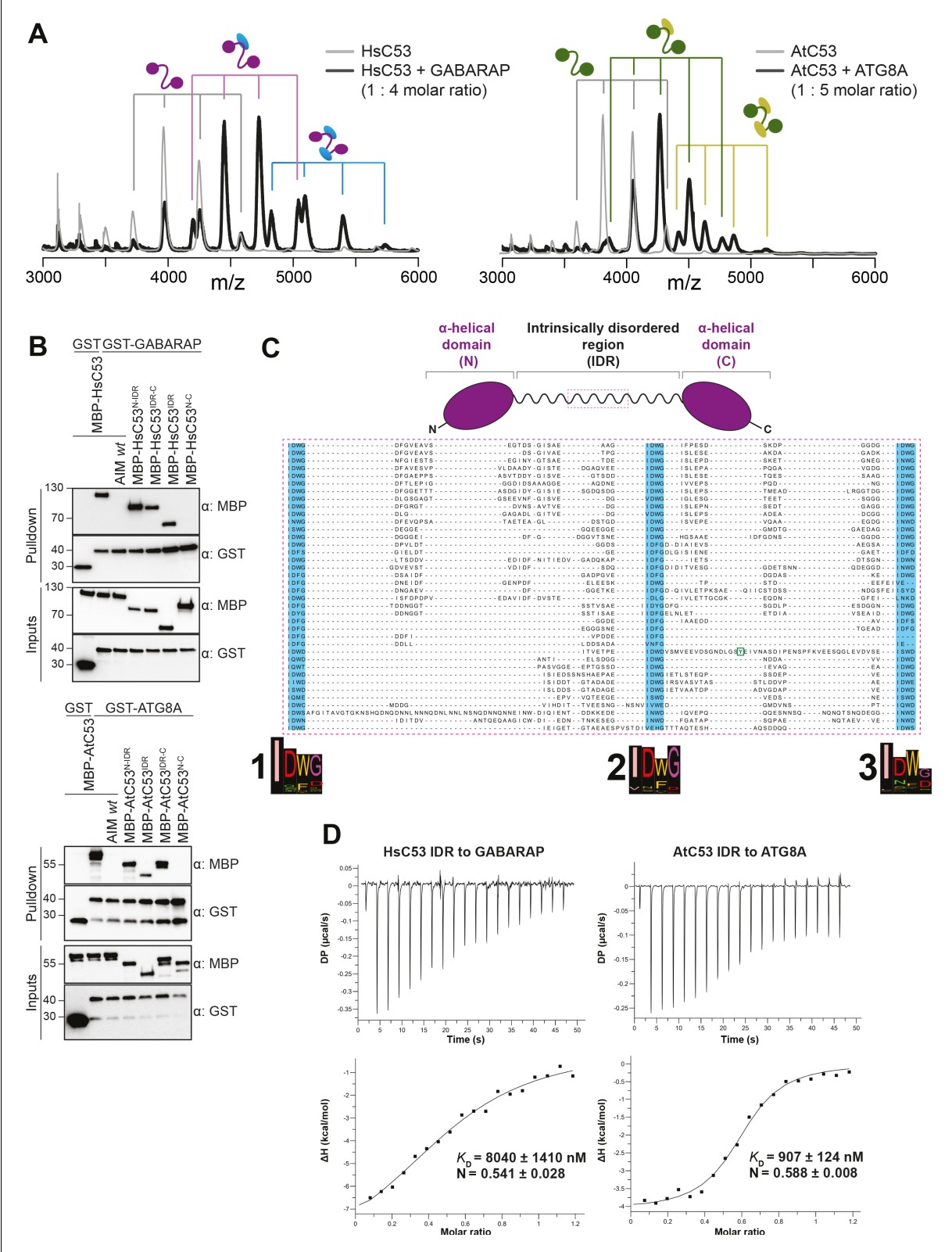

**Figure 4.** C53 interacts with ATG8 via the Intrinsically Disordered Region. (**A**) Native mass spectrometry (nMS) analysis showing HsC53 and AtC53 form 1:1 and 1:2 complexes with GABARAP and ATG8A, respectively. Left; nMS of HsC53 (grey) and HsC53 plus GABARAP in a 1:4 molar ratio (black). Peaks corresponding to unbound HsC53, the 1:1 complex and 1:2 complex are indicated with grey, magenta and blue, respectively. Right; nMS of AtC53 (grey) and AtC53 plus ATG8A in a 1:5 molar ratio (black). Peaks corresponding to unbound AtC53, the 1:1 complex and 1:2 complex are indicated with

*Figure 4 continued on next page*

*Figure 4 continued*

grey, green and yellow, respectively. Full spectra are shown in *Figure 4—figure supplement 1*. (B) *Upper Panel,* HsC53 intrinsically disordered region (IDR) is necessary and sufficient to mediate the interaction with GABARAP. *Lower Panel,* AtC53 IDR is necessary and sufficient to mediate the interaction with ATG8A. Bacterial lysates containing recombinant protein were mixed and pulled down with glutathione magnetic agarose beads. The AIM *wt* peptide was added at a final concentration of 200 µM. Input and bound proteins were visualized by immunoblotting with anti-GST and anti-MBP antibodies. N: N-terminal truncation; M: IDR; C: C-terminal truncation. (C) C53 IDR has three highly conserved regions. Protein sequence alignment of the predicted IDR amino acid sequences showed three highly conserved regions with a consensus sequence of IDWG (highlighted in blue). Y304 is highlighted in the green rectangle. The species names and the full protein sequence alignment is presented in Figure S10. (D) Isothermal titration calorimetry (ITC) experiments showing binding of AtC53 and HsC53 IDRs to ATG8A and GABARAP, respectively. Upper left and right panels show heat generated upon titration of AtC53 IDR (250 µM) or HsC53 IDR (250 µM) to ATG8A or GABARAP (both 40 µM). Lower left and right panels show integrated heat data (■) and the fit (solid line) to a one-set-of-sites binding model using PEAQ-ITC analysis software. Representative values of $K_D$, N, ΔH, -TΔS, and ΔG from three independent ITC experiments are reported in *Supplementary file 6*.

The online version of this article includes the following figure supplement(s) for figure 4:

**Figure supplement 1.** AtC53-ATG8 interaction is not mediated by canonical ATG8 interaction motifs.
**Figure supplement 2.** Native mass spectrometry analyses of HsC53-GABARAP and AtC53-ATG8A interactions.
**Figure supplement 3.** Multiple sequence alignment of C53 homologs.

DDRGK1 with AtC53 in punctate structures increases upon ER stress and these puncta are delivered to the vacuole (*Figure 8C,D*, *Figure 8—figure supplement 1C,D*). Strikingly, AtC53 autophagic flux requires functional UFL1 and DDRGK1, as the number of AtC53 puncta was significantly lower in ufl1 and ddrgk1 mutants (*Figure 8E*, *Figure 8—figure supplement 1E*). Ultimately, autophagic flux assays using the ufmylation machinery mutants confirmed that AtC53 flux requires a functional ufmylation machinery (*Figure 8E*, *Figure 8—figure supplement 1F,G*). Taken together, our data indicate that C53 is recruited to the ER by forming a heteromeric receptor complex with UFL1 and DDRGK1.

Since, DDRGK1 is an ER-membrane protein and physically linked to C53, we analyzed the degradation of DDRGK1 during ER stress. Transgenic lines expressing DDRGK1-GFP in *c53* and *atg5* mutant revealed that recruitment of DDRGK1 from ER membrane to punctate structures during ER stress required both C53 and ATG5 (*Figure 8—figure supplement 2A*). Furthermore, DDRGK1 puncta colocalized with mCherry-ATG8A in a C53-dependent manner (*Figure 8—figure supplement 2B*). Western-blot-based autophagic flux assays further confirmed AtC53-dependent degradation of DDRGK1 (*Figure 8—figure supplement 2C*). Interestingly, abundant ER proteins such as the Calnexin, BIP or SMT1 are not degraded by AtC53-dependent ER-phagy. Likewise, small and large ribosomal subunits are not degraded by AtC53 (*Figure 8—figure supplement 2C*). These results are consistent with the C53 cargo clientele defined by quantitative proteomics, and point toward a highly selective, yet unknown cargo selection mechanism of C53.

We then explored how C53 is kept inactive under normal conditions. We hypothesized that the Ubiquitin like modifier UFM1 may safeguard the C53 receptor complex under normal conditions and keep ATG8 at bay. Upon ER stress, UFM1 would be transferred to RPL26, exposing sAIMs on C53. To test this, we first analyzed the UFM1-C53 interaction by in vitro pull-down assays and could show that AtC53 can interact with AtUFM1 (*Figure 9A*). Furthermore, in vitro competition experiments revealed a competition between UFM1 and ATG8 for C53 binding (*Figure 9A*). This result is reminiscent of the mutually exclusive UFM1 and GABARAP binding of UBA5, the E1 enzyme in the ufmylation cascade (*Huber et al., 2019*). We then performed in vivo co-immunoprecipitation experiments during ER stress. Consistent with our hypothesis and in vitro data, ER stress led to depletion of UFM1 and enhanced AtC53-ATG8 interaction (*Figure 9B,C*, supplement 1). Altogether, these data suggest that the two ubiquitin-like proteins UFM1 and ATG8 compete with each other for association with the C53 receptor complex (*Figure 9D*).

## C53 is crucial for ER stress tolerance

Finally, we examined if C53 is physiologically important for ER stress tolerance. First, we tested if C53 plays a general role in autophagy using carbon and nitrogen starvation assays. Carbon and nitrogen starvation are typically used to characterize defects in bulk autophagy responses (*Marshall and Vierstra, 2018*). In contrast to the core autophagy mutants *atg5* and *atg2*, CRISPR-generated *Atc53* mutants did not show any phenotype under carbon or nitrogen starvation conditions (*Figure 10A,B*). However, consistent with increased flux, *Atc53* mutants were highly sensitive

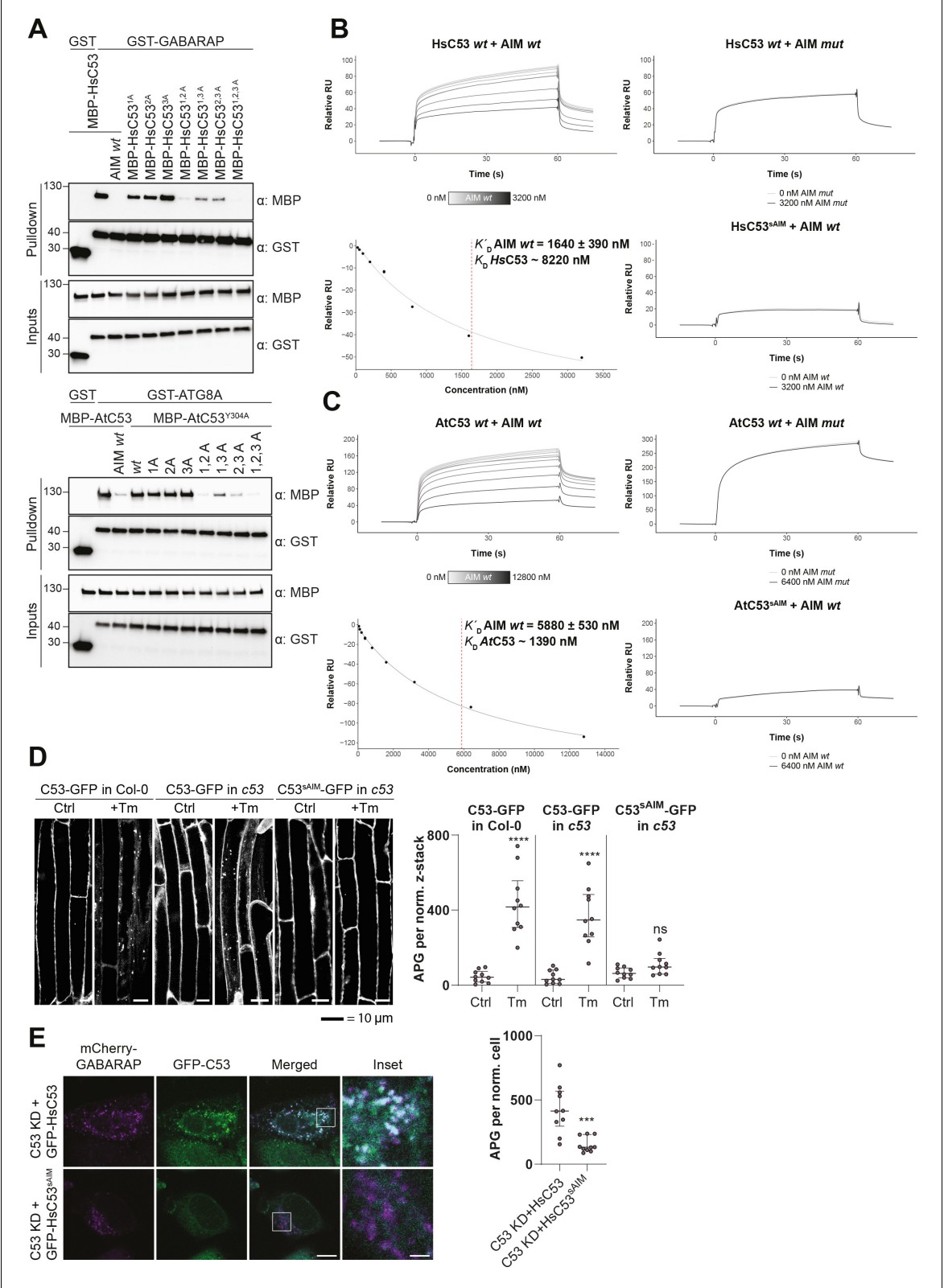

**Figure 5.** C53 interacts with ATG8 via shuffled ATG8 interacting motifs (sAIMs). (**A**) *Upper Panel*, the three conserved IDWG motifs (sAIMs) in *Hs*C53 IDR mediate interaction with GABARAP. Pull downs were performed as described in (**b**). 1A: W269A; 2A: W294A; 3A: W312A. *Lower Panel*, *At*C53 quadruple mutant cannot interact with ATG8A. In addition to the sAIM motifs, a canonical AIM (304-YEIV) also contributes to ATG8 binding. Bacterial lysates containing recombinant protein were mixed and pulled down with glutathione magnetic agarose beads. The AIM *wt* peptide was added at a

*Figure 5 continued on next page*

*Figure 5 continued*

final concentration of 200 µM. Input and bound proteins were visualized by immunoblotting with anti-GST and anti-MBP antibodies. 1A: W276A; 2A: W287A; 3A: W335A. (**B, C**) Surface plasmon resonance (SPR) analyses of C53-ATG8 binding. GST-GABARAP or GST-ATG8A fusion proteins were captured on the surface of the active cell (500 RU) and GST alone was captured on the surface of the reference cell (300 RU). *Upper Left Panels:* Increasing concentrations of the AIM *wt* peptide were premixed with 10 µM C53 and injected onto the sensor surface. Binding curves were obtained by subtracting the reference cell signal from the active cell signal. *Lower left Panels*: Binding affinities were determined by plotting the maximum response units versus the respective concentration of the AIM *wt* peptide and the data were fitted using the Biacore T200 Evaluation software 3.1 (GE Healthcare). *Upper Right Panels:* C53 was premixed with buffer or 3600 nM of AIM *mut* peptide and injected onto the sensor surface. *Lower Right Panels:* C53$^{sAIM}$ was premixed with buffer or 3600 nM of AIM wt peptide and injected onto the sensor surface. A representative sensorgram from three independent experiments is shown. (**D**) *At*C53 quadruple mutant (sAIM) does not form puncta upon ER-stress. *Left Panel*, representative confocal images of transgenic Arabidopsis seedlings expressing AtC53-GFP or AtC53$^{sAIM}$-GFP in Col-0 wild type and *c53* mutant backgrounds. Four-day-old seedlings were incubated in either control or tunicamycin (10 µg/ml) containing media. Scale bars, 10 µm. *Right Panel*, Quantification of autophagosomes (APG) per normalized Z-stacks. Bars represent the mean (± SD) of at least 10 biological replicates. (**E**) *Hs*C53 sAIM mutant does not form puncta upon ER-stress. *Left Panel,* Confocal images of PFA fixed C53 knockdown HeLa cells transiently expressing *Hs*C53-GFP or *Hs*C53$^{sAIM}$-GFP (green) and mCherry-GABARAP (magenta). Cells were treated for 16 hr with 2.5 µg/ml tunicamycin (Tm). Scale bar, 10 µm. Inset scale bar, 2 µm. Representative images are shown. *Right Panel,* Quantification of autophagosomes (APG) per normalized cell. Bars represent the mean (± SD) of at least 10 biological replicates. Significant differences are indicated with * when p value ≤ 0.05, ** when p value ≤ 0.01, and *** when p value ≤ 0.001.

The online version of this article includes the following figure supplement(s) for figure 5:

**Figure supplement 1.** Biophysical characterization of sAIM mediated C53-ATG8 interaction.

to phosphate starvation, which has been shown to trigger an ER stress response (***Naumann et al., 2019***; ***Figure 10C***, ***Figure 10—figure supplement 1A***). Similarly, in both root length and survival assays, *Atc53* mutants were sensitive to tunicamycin treatment (***Figure 10D***, ***Figure 10—figure supplement 1B,C***). In addition, ufmylation machinery mutants (***Figure 10E***), including *ufl1* and *ddrgk1*, were also sensitive to tunicamycin treatment but insensitive to carbon and nitrogen starvation (***Figure 10F***, ***Figure 10—figure supplement 1D,E***). Lastly, the *Marchantia polymorpha c53* mutant was also sensitive to tunicamycin, suggesting C53 function is conserved across the plant kingdom (***Figure 10—figure supplement 1F***). We then performed complementation assays using wild-type AtC53 and the AtC53$^{sAIM}$ mutant. AtC53 expressing lines behaved like wild-type plants in tunicamycin supplemented plates (***Figure 10G***). However, AtC53$^{sAIM}$ mutant did not complement the tunicamycin sensitivity phenotype, and had significantly shorter roots (***Figure 10G***, ***Figure 10—figure supplement 1G***). Parallel to analyzing C53-mediated ER homeostasis in plants, stress tolerance assays in HeLa cells showed that silencing of HsC53 led to an induction of Bip3 chaperone protein levels (***Figure 10—figure supplement 1H***), indicating increased ER stress. Complementation of *Hsc53* silenced lines with HsC53-GFP dampened Bip3 expression (***Figure 10—figure supplement 1H***). Altogether, these results demonstrate that C53 coordinated ER-phagy is crucial for ER stress tolerance in plant and mammalian cells.

## Discussion

The endoplasmic reticulum is a highly heterogeneous and dynamic network that handles folding and maturation of up to a million proteins per minute in a cell (***Karagöz et al., 2019***). It constantly tailors the proteome in a cell-type and physiological state dependent manner. Unfortunately, protein synthesis, folding, and maturation events are all error prone, and faulty proteins have to be efficiently recycled to prevent accumulation of toxic by-products. Since, selective autophagy is a highly efficient quality control pathway that could very quickly recycle large amounts of proteins and membranous compartments, it is not surprising to have various ER-phagy receptors focusing on re-shaping the ER during stress (***Chino and Mizushima, 2020***; ***Wilkinson, 2020***). With C53, eight ER-phagy receptors have now been identified in metazoans, working together to maintain ER homeostasis under changing cellular conditions. However, since most of ER-phagy pathways were studied during nutrient starvation, which supposedly activates bulk recycling mechanisms, selective cargo recruitment triggered upon quality control defects is still poorly understood. It is thus a major challenge to elucidate the coordination of different ER-phagy receptors and their cross-talk with the core ER-quality control pathways (***Chino and Mizushima, 2020***).

Our findings reveal C53-mediated ER-phagy to be a central mechanism operating at the interface of key quality control pathways, controlling ER homeostasis across different kingdoms of life. Using

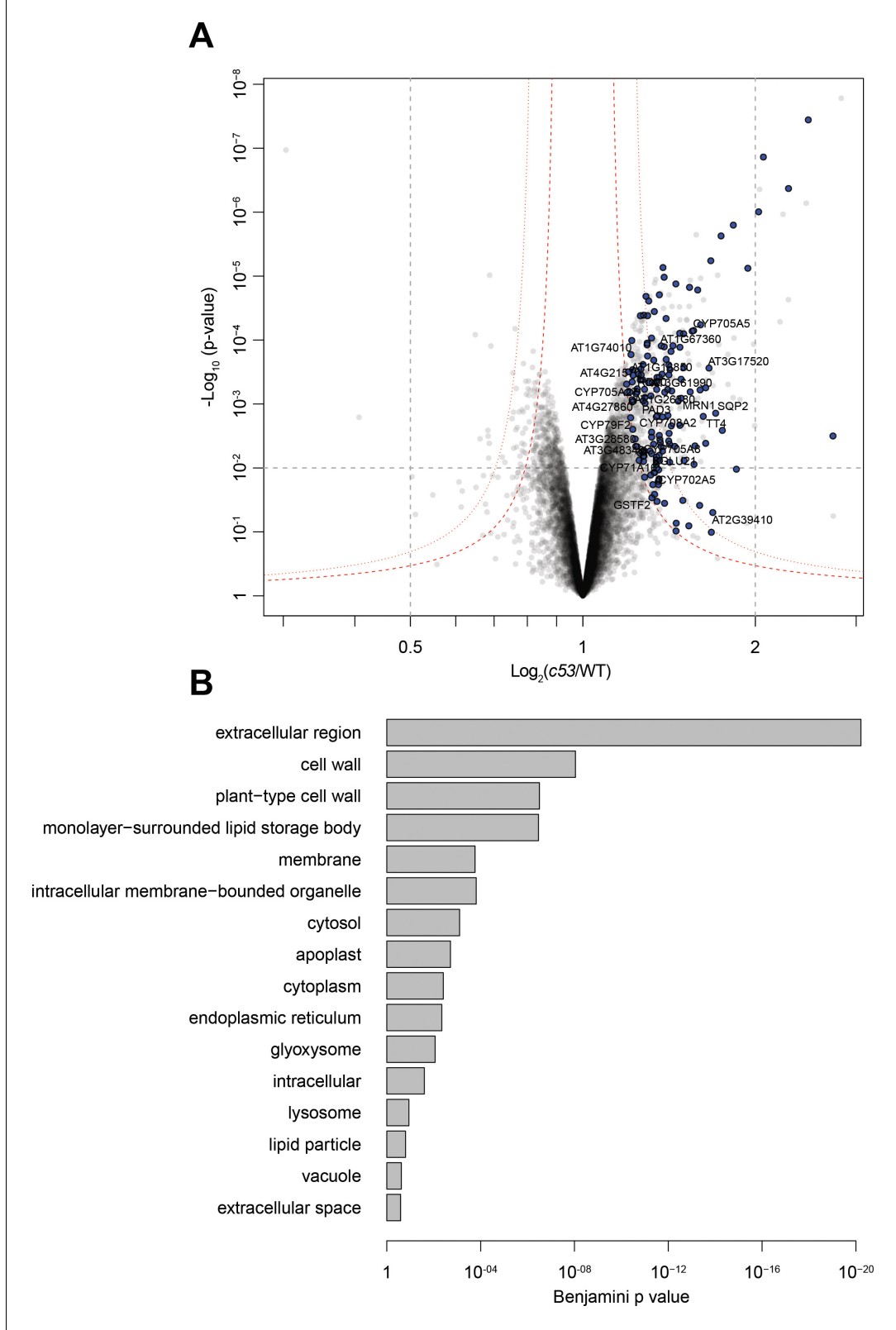

**Figure 6.** Quantitative proteomics analyses of AtC53 mediated degradation. (**A**) Volcano plot showing proteins that are accumulating in *Atc53 mutants* Names of ER resident proteins are shown. Proteins that are labeled with blue either reside or mature at the ER. (**B**) GO analysis of proteins accumulating in *Atc53*. See ***Supplementary file 3*** and ***4*** for details.

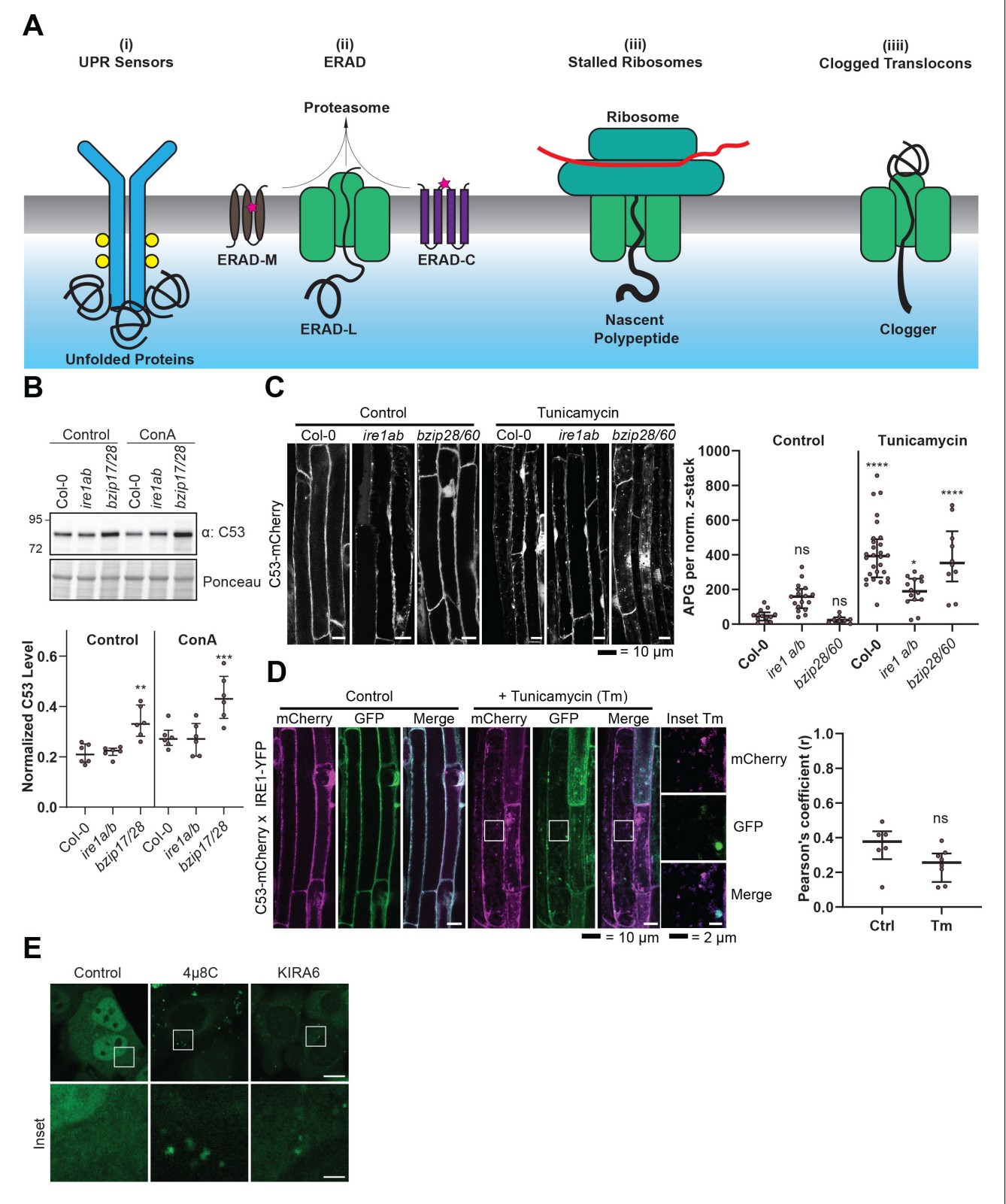

**Figure 7.** C53 is not activated by associating with UPR sensors. (**A**) Cartoon depicting the four scenarios we tested to understand the mechanism of activation of C53. (**B**) AtC53 flux is enhanced in Arabidopsis UPR sensor mutants. *Upper Panel*, representative western blot image of autophagic flux analysis of C53 in Col-0 wild type, *ire1a/b*, and *bzip17/28* double mutants. Seedlings were incubated in either control (Ctrl) or 1 μM concanamycin A (ConA) containing medium for 16 hr. Proteins extracted from whole seedlings were analysed by immunoblotting with anti-C53 antibody. Total proteins

*Figure 7 continued on next page*

*Figure 7 continued*

were analysed by Ponceau S staining. *Lower Panel*, Quantification of the intensities of the C53 bands normalized to the total protein level of the lysate (Ponceau S). Average C53 levels and SD for n = 6 are shown. Significant differences are indicated with * when p value ≤ 0.05, ** when p value ≤ 0.01, and *** when p value ≤ 0.001. (**C**) AtC53 still form puncta upon ER stress in *ire1a/b* and *bzip28/60* double mutants. *Left Panel,* Confocal micrographs of autophagic flux of transgenic seedlings expressing C53-mCherry in Col-0 wild type, the *ire1a/b* and *bzip28/60* double mutant. Seedlings were incubated in either control media or media containing 10 µg/ml tunicamycin (Tm). Representative confocal images in single plane are shown. Scale bars, 10 µm. *Right Panel,* Quantification of autophagosomes (APG) per normalized Z-stack. Bars represent the mean (± SD) of at least 10 biological replicates (**D**) AtC53 does not colocalize with IRE1-YFP during ER stress. *Left Panel*, Co-localization analyses of single plane confocal images obtained from transgenic Arabidopsis roots co-expressing AtC53-mCherry (magenta) and IRE1-YFP (green) in wild type Col-0 background. Four-day-old seedlings were incubated in either control or 10 µg/ml tunicamycin containing media. Scale bars, 20 µm. Inset scale bars, 2 µm. *Right Panel*, Pearson's Coefficient (r) analysis of the colocalization of AtC53-mCherry and IRE1-YFP. Bars represent the mean (± SD) of at least five biological replicates (**E**) Chemical inhibition of IRE1 activity enhances HsC53 autophagic flux in HeLa cells. Confocal micrographs of PFA fixed HeLa cells transiently expressing HsC53-GFP. Cells were either left untreated (Control) or treated for 1 hr with 5 µM 4µ8C (IRE1 RNase activity inhibitor) or 1 µM KIRA6 (IRE1 kinase activity inhibitor). Inhibitor treatments led to the depletion of HsC53 from the nucleus and puncta formation. Scale Bar, 20 µm. Representative images are shown.

The online version of this article includes the following figure supplement(s) for figure 7:

**Figure supplement 1.** C53 is not activated by model ERAD substrates.

**Figure supplement 2.** C53 is activated upon ribosome stalling during co-translational protein translocation.

various model systems including divergent model plant species and human cell lines, we show that C53 forms an ancient autophagy receptor complex that is closely connected to the ER quality control system via the ufmylation pathway. Unlike other ER-phagy receptors studied so far, C53 seem to be highly specific in resolving ribosome stalling triggered during SRP-dependent co-translational protein translocation. However, it remains to be shown how C53 recruit specific cargo into the autophagosomes.

Interestingly, recent genome wide CRISPR screens identified ufmylation as a major regulator of ER-phagy, the ERAD pathway, and viral infection (*Leto et al., 2019*; *Liang et al., 2020*; *Kulsuptrakul et al., 2019*). Using fluorescent reporter lines and genome wide CRISPRi screens, Liang et al., showed that ufmylation plays a major role in regulating starvation induced ER-phagy. They showed that both DDRGK1 and UFL1 are critical for starvation-induced ER-phagy, whereas C53 mutants did not show any ER-phagy defects (*Liang et al., 2020*). Our results using stable transgenic organisms show that C53-mediated autophagy is not activated by carbon or nitrogen starvation that are typically used to activate bulk autophagy (*Figure 2* and *Figure 2—figure supplement 1*). C53 is activated by ER stress caused by phosphate depletion (*Naumann et al., 2019*). Consistently, phenotyping experiments revealed that C53 and the ufmylation machinery mutants are asymptomatic during carbon or nitrogen starvation but are highly sensitive to ER-stress treatments (*Figure 10*, *Figure 10—figure supplement 1*). Together, the two complementary studies indicate that the ufmylation machinery is tightly associated with ER-phagy in multicellular eukaryotes and plays a crucial role in ER stress tolerance.

It should be noted that C53 and ufmylation proteins are essential for mammalian development (*Gerakis et al., 2019*). Defects in C53 receptor complex have been associated with various diseases including liver cancer, pancreatitis, and cardiomyopathy (*Gerakis et al., 2019*). Our results suggest C53 and ufmylation is also critical for stress tolerance in plants, but they are not essential for development; suggesting plants have evolved compensatory mechanisms during adaptation to sessile life. Future comparative studies could reveal these mechanisms and help us develop sustainable strategies for promoting ER proteostasis during stress in mammals and plants.

## Materials and methods

### Cloning procedures

Constructs for *Arabidopsis thaliana* and *E. coli* transformation were generated using the GreenGate (GG) cloning method (*Lampropoulos et al., 2013*). Plasmids used are listed in materials section. The coding sequence of genes of interest were either ordered from Twist Biosciences or Genewiz or amplified from Col-0 or HeLa cDNA using the primers listed in the materials section. The internal BsaI sites were mutated by site-directed-mutagenesis without affecting the amino acid sequence.

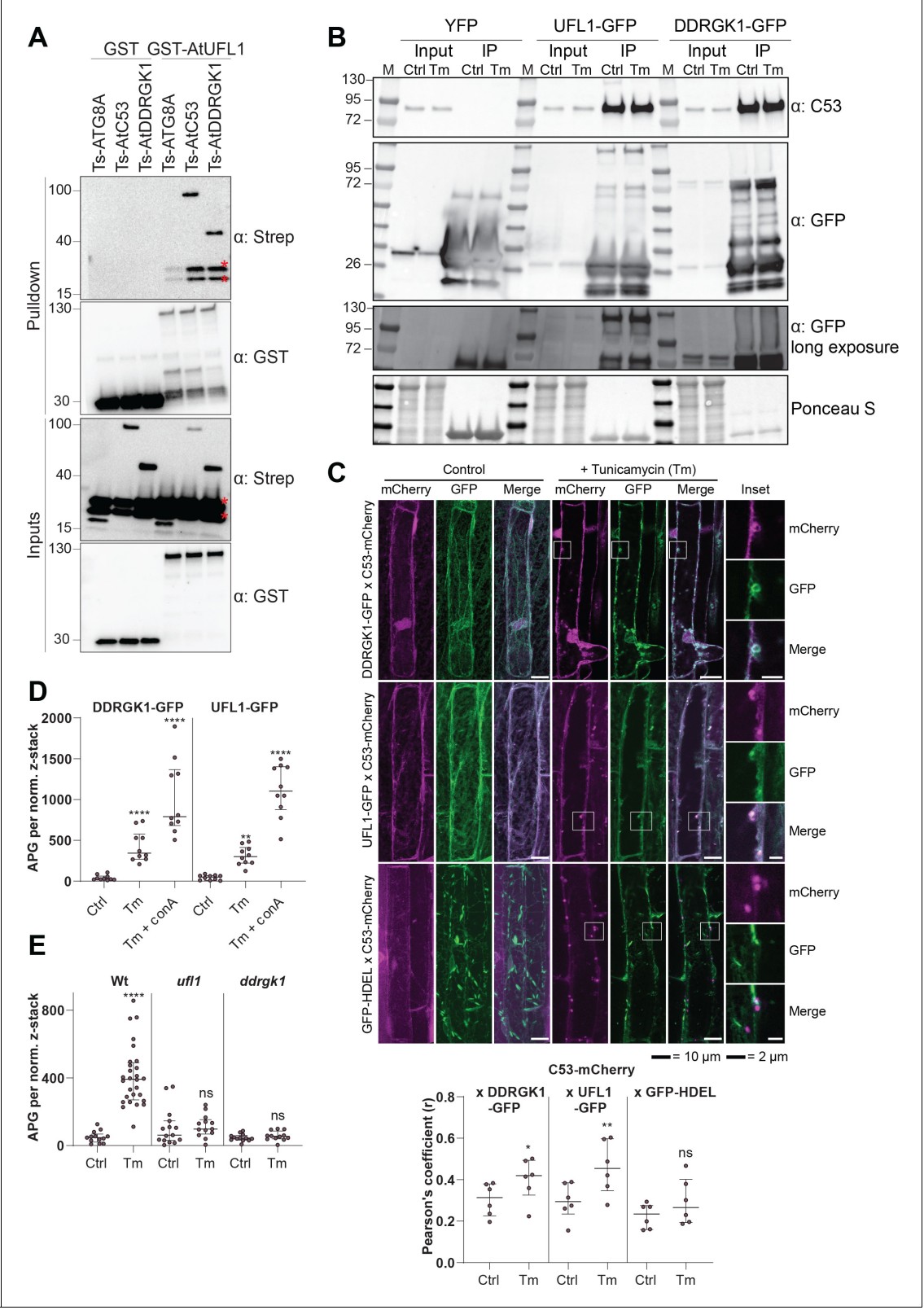

**Figure 8.** C53 forms a heteromeric receptor complex with UFL1 and DDRGK1. (**A**) AtUFL1 interacts with AtC53 and AtDDRGK1. Bacterial lysates containing recombinant protein were mixed and pulled down with glutathione magnetic agarose beads. Input and bound proteins were visualized by immunoblotting with anti-GST and anti-Strep antibodies. Red asterisks indicate endogenous *E. coli* biotinylated proteins. (**B**) AtC53 associates with AtUFL1 and AtDDRGK1. In vivo co-immunoprecipitation of UFL1-GFP or DDRGK1-GFP expressing Arabidopsis seedlings incubated in either control

*Figure 8 continued on next page*

*Figure 8 continued*

(Ctrl) or 10 µg/ml tunicamycin (Tm) containing media. (**C**) AtDDRGK1 and AtUFL1 colocalize with AtC53 puncta upon ER stress induction. *Upper Panel*, Co-localization analyses of confocal micrographs of wild type Col-0 roots co-expressing AtC53-mCherry (magenta) with DDRGK1-GFP, UFL1-GFP, or GFP-HDEL (green). Transgenic seedlings were incubated in either control or tunicamycin (10 µg/ml) containing media. Representative confocal images of control conditions are shown in maximum projection to emphasize ER association. Images of tunicamycin treatments are shown in single plane. Scale bars, 10 µm. Inset scale bars, 2 µm. *Lower Panel*, Pearson's Coefficient colocalization analysis per normalized Z-scan. Bars represent the mean (± SD) of 5 biological replicates. (**D**) DDRGK1 and UFL1 undergo vacuolar degradation upon ER stress induction. Quantification of confocal micrographs of autophagic flux of UFL1-GFP and DDRGK1-GFP. Seedlings were incubated in either control, 10 µg/ml tunicamycin (Tm), or 10 µg/ml tunicamycin with 1 µM Concanamycin A (Tm+ConA) media. Quantification of autophagosomes (APG) per normalized Z-stacks of UFL1-GFP and DDRGK1-GFP. Bars represent the mean (± SD) of at least 10 biological replicates. (**E**) AtC53 vacuolar degradation requires DDRGK1 and UFL1. Quantification of confocal images of wild type (Col-0), *ufl1*, and *ddrgk1* Arabidopsis seedlings expressing AtC53-mCherry. Six-day-old seedlings were incubated in either control (Ctrl) or 10 µg/ml tunicamycin (Tm) containing media. Scale bars, 20 µm. Quantification of autophagosomes (APG) per normalized Z-stacks. Bars represent the mean (± SD) of at least 10 biological replicates. Significant differences compared to control treatment (Ctrl) are indicated with * when p value ≤ 0.05, ** when p value ≤ 0.01, and *** when p value ≤ 0.001.

The online version of this article includes the following figure supplement(s) for figure 8:

**Figure supplement 1.** C53, DDRGK1, and UFL1 form a heteromeric receptor complex.
**Figure supplement 2.** AtC53 mediates autophagic degradation of DDRGK1.

For *Marchantia polymorpha* Gateway Cloning (*Ishizaki et al., 2015*) was used to generate all constructs.

For HeLa expression experiments, plasmids used are listed in the materials section. The constructs were made by conventional restriction enzyme-based cloning.

## CRISPR/Cas9 construct design

The CRISPR/Cas9 constructs for mutating *c53*, *DDRGK1* and *UFM1* in *Arabidopsis thaliana* were prepared according to the protocol described by *Xing et al., 2014* and *Ma et al., 2015*. The pHEE401E and pCBC-DT1T2 vectors for expressing two sgRNAs were provided by Youssef Belkhadir and Jixiang Kong, GMI Vienna. sgRNA target sites were chosen using the website http://crispr.hzau.edu.cn/CRISPR2/. Each gene was targeted by two sgRNAs to remove a fragment of the gene. The CRISPR cassettes of each gene were generated by PCR amplification using pCBC-DT1T2 as template with the primer pairs BsF/F0 and BsR/R0, using adaptors containing the BsaI-restriction sites, respectively (see materials section). The PCR products were digested with *Bsa*I, ligated into the pHEE401E plasmid, and transformed into DH5α *E. coli*. Floral dipping proceeded as described previously (*Clough and Bent, 1998*). Genotyping primers P1 5′-xxx-3′ and P2 5′-xxx-3′ flanking each target site were used to select T1 plants that carried deletions. Sanger sequencing was performed to define the deletion. Through backcrossing with Col-0 plants and genotyping, Cas9-free plants were achieved.

In *Marchantia polymorpha*, CRISPR/Cas9 constructs were generated by selecting two target sequences in *c53* and *ire1*. Synthetic oligonucleotides were annealed and inserted at the BsaI site of the entry vector pMpGE_En03 flanked by attL1 and attL2 sequences (*Sugano et al., 2018*). The resultant cassettes were inserted to the destination vector pMpGE011 by LR Clonase II Enzyme Mix. The vectors were introduced into thalli of TAK1 via *A. tumefaciens* GV3101+pSoup, and the transformants were selected with 0.5 µM chlorsulfuron (*Kubota et al., 2013*). Genomic DNA from transformants was amplified by PCR and sent for sequencing to verify mutations.

## Plant materials and growth conditions

All *Arabidopsis thaliana* lines used originate from the Columbia (Col-0) ecotype background. Mutant lines used in this study are listed in the materials section. All transgenic plants were generated by the floral dipping method (*Clough and Bent, 1998*) for which the plasmid constructs were prepared using the green gate cloning method (*Lampropoulos et al., 2013*).

Seeds were then spread on plates or liquid culture with half-strength MS media (Murashige and Skoog salt + Gamborg B5 vitamin mixture) with 1% sucrose, 0.5 g/L MES and 1% plant agar. pH was adjusted to 5.7 with NaOH. Seeds were imbibed for 4 days at 4°C in darkness. Plants were grown at 21°C at 60% humidity under LEDs with 50 µM/m$^2$s and 12 hr:12 hr photoperiod.

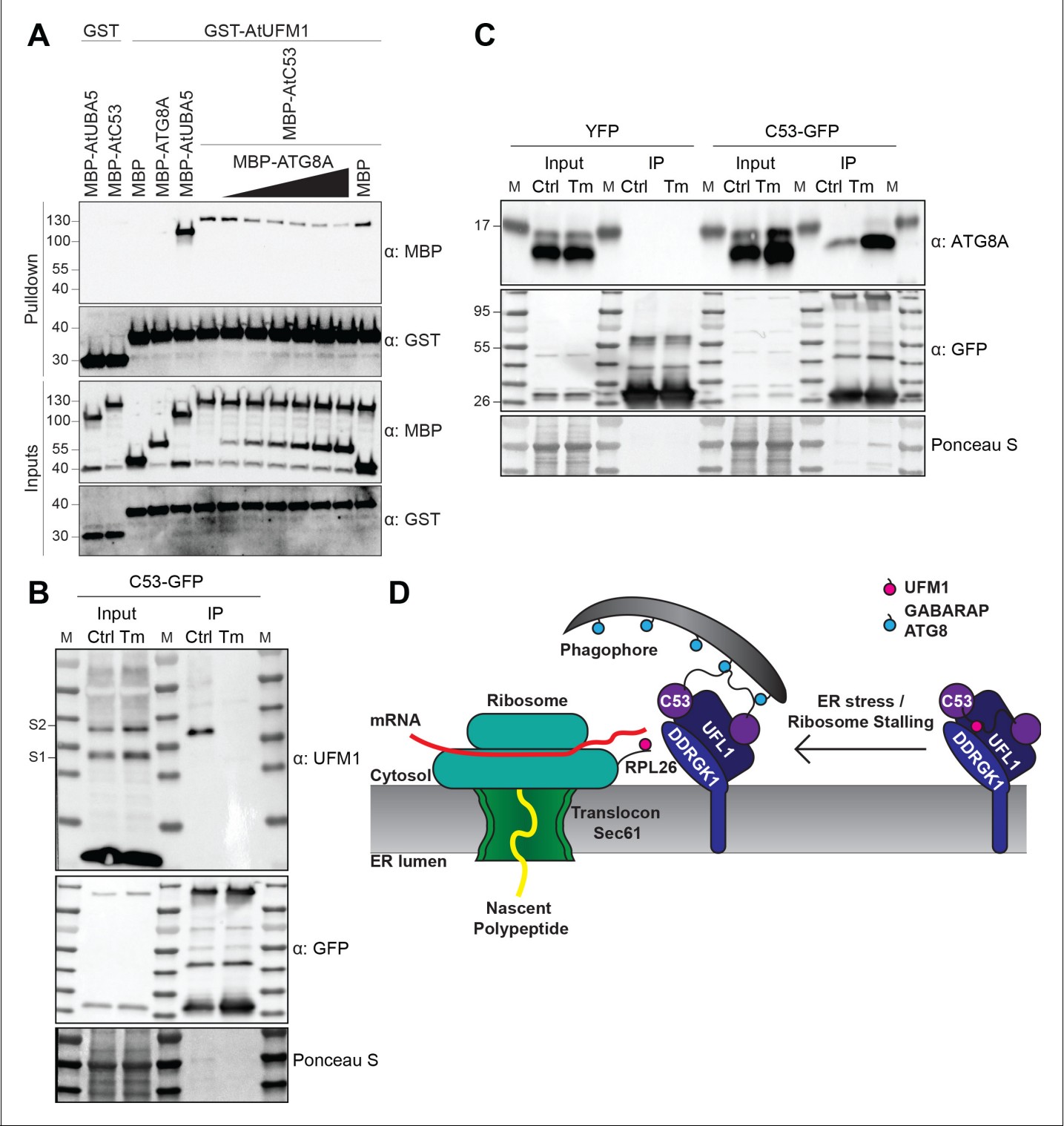

**Figure 9.** C53 autophagic flux is activated by depletion of UFM1 during ER stress. (**A**) AtC53 directly interacts with UFM1 and this interaction becomes weaker upon increasing concentrations of ATG8A. Bacterial lysates containing recombinant protein were mixed and pulled down with glutathione magnetic agarose beads. Input and bound proteins were visualized by immunoblotting with anti-GST and anti-MBP antibodies. The red asterisk indicates MBP-ATG8A. (**B**) AtC53-ATG8 association becomes stronger upon ER stress induction triggered by tunicamycin. In vivo co-immunoprecipitation of extracts of Arabidopsis seedlings expressing AtC53-GFP incubated in either control (Ctrl) or 10 μg/ml tunicamycin (Tm) containing media. (**C**) AtC53-UFM1 association becomes weaker upon ER stress induction triggered by tunicamycin. In vivo pull downs of extracts of Arabidopsis seedlings expressing C53-GFP incubated in either control (Ctrl) or 10 μg/ml tunicamycin (Tm) containing media. Input and bound proteins

*Figure 9 continued on next page*

*Figure 9 continued*
were visualized by immunoblotting with the indicated antibodies. (D) Current working model of the C53 receptor complex. Upon ribosome stalling, UFM1 is transferred from the C53 receptor complex to the tail of RPL26, exposing the sAIMs on C53 for ATG8 binding and subsequent recruitment to the autophagosomes.

For the autophagy flux assay, TMT and in vivo immunoprecipitation, seedlings were grown in liquid culture under continuous light.

Male *Marchantia polymorpha* accession Takaragaike-1 (Tak-1) was maintained asexually and cultured through gemma using half-strength Gamborg's B5 medium containing 1% agar under 50–60 mmol photons $m^{-2}s^{-1}$ continuous white light at 22° C unless otherwise defined (*Kubota et al., 2013*).

## Plant sensitivity tests
### Arabidopsis thaliana
#### Root-length quantification
Seedlings were grown for 9 days on media supplemented with the indicated drug concentration. Plates were scanned on day 0 and then quantified daily starting from day 2 to day 9. Large-scale root-length quantification was conducted using the automated plant imaging analysis software BRAT (Buschlab Root Analysis Toolchain) (*Slovak et al., 2014*) with the inhouse high-performance computer cluster MENDEL. Before analysis, collected data was passed through software quality control.

## Starvation treatments
Carbon starvation: Seedlings were grown on half-strength MS media with 1% sucrose for 7 days. They were then transferred to media without sucrose, followed by wrapping the plates in aluminium foil and placing them under the same growth conditions as before for 9 days.

Nitrogen starvation: Seedlings grew on half-strength MS media with 0.5% sucrose for 7 days. They were then transferred to media without nitrogen and put under the same growth conditions as before for 14 days.

Seedlings were arranged in a similar fashion to *Jia et al., 2019*.

Phosphate starvation: The method was previously described by *Naumann et al., 2019*. Seeds were surface-sterilized and germinated 5 days on +Pi medium prior to transfer to 1% (w/v) Phyto-Agar (Duchefa) containing 2.5 mM $KH_2PO_4$, pH 5.6 (high Pi or +Pi medium) or no Pi supplement (low Pi or –Pi medium), 5 mM $KNO_3$, 0.025 mM Fe-EDTA, 2 mM $MgSO_4$, 2 mM Ca(NO3)2, 2.5 mM MES-KOH, 0.07 mM H3BO3, 0.014 mM $MnCl_2$, 0.01 mM NaCl, 0.5 µM $CuSO_4$, 1 µM $ZnSO_4$, 0.2 µM $Na_2MoO_4$, 0.01 µM $CoCl_2$, 5 g/L sucrose. The agar was routinely purified by repeated washing in deionized water and underwent subsequent dialysis using DOWEX G-55 anion exchanger (*Ticconi et al., 2009*). ICP-MS analysis of the treated agar (7.3 µg/g Fe and 5.9 µg/g P) indicated a contribution of 1.25 µM Fe and 1.875 µM P to the solid 1% agar medium. Images were analyzed using ImageJ software.

## Survival assay
Seedlings were grown on 9 cm round plates supplemented with the indicated drug at the indicated concentration. Seedling survival was quantified after 14 days. Differentiation between live and dead seedlings was carried out similar to *Yang et al., 2016*. Surviving seedlings were defined as seedlings which had two green cotyledons and two green true leaves. Plants with yellow leaves or cotyledons were defined as dead.

## Marchantia polymorpha
Tunicamycin sensitivity: 14 days old plants were transformed to half-strength Gamborg's B5 medium containing indicated concentration of tunicamycin and grown in continues light at 22℃ to determine survival rates.

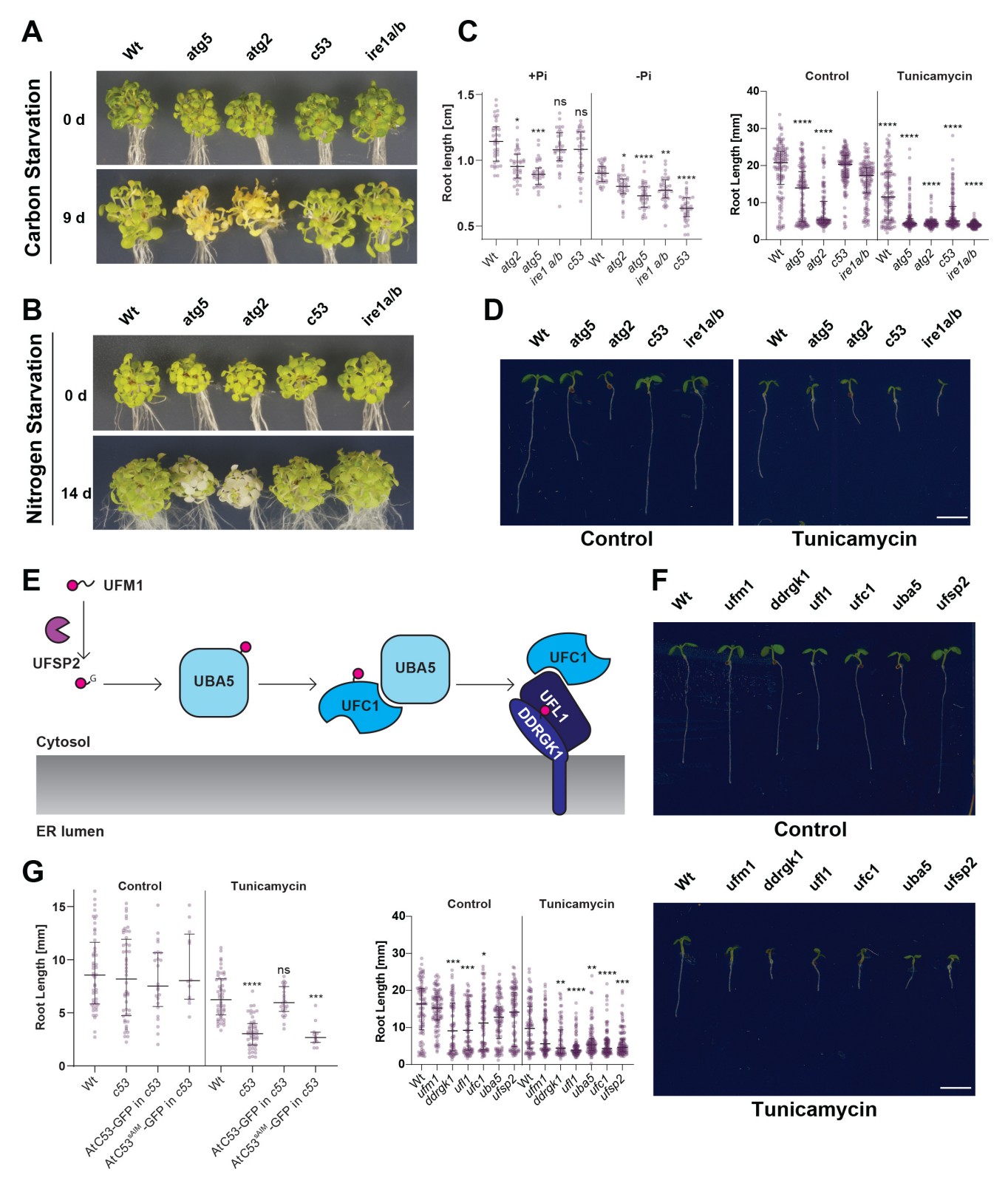

**Figure 10.** C53 is crucial for ER stress tolerance. (**A**) *Atc53* mutant is insensitive to carbon starvation. Phenotypes before (0 d) and after 9 days carbon starvation (9 d) of 7-day-old seedlings, n ≥ 20 seedlings per genotype. (**B**) *Atc53* mutant is insensitive to nitrogen starvation. Phenotypes before (0 d) and after 14 days nitrogen starvation (14 d) of 7-day-old seedlings, n ≥ 20 seedlings per genotype. (**C**) *Atc53* mutant is sensitive to phosphate starvation. Root-length quantification of seven-day-old seedlings which were transferred to media with or without Pi supplement (+Pi, -Pi), and imaged

*Figure 10 continued on next page*

*Figure 10 continued*

after 2 days. (D) Atc53 mutants are sensitive to ER stress induced by tunicamycin. Root-length quantification of 7-day-old seedlings grown on half strength MS media without sucrose treated with 100 ng/mL tunicamycin (Tm). Bottom Panel, Root-length quantification of 7-day-old seedlings. n ≈ 125 seedlings per genotype and treatment. *Left Panel*, Example of 7-day-old seedlings grown in described conditions. Scale bars = 5 mm. Left, non-treated seedlings. Right, seedlings grown at 100 ng/mL Tm. *Right Panel*, Root length of each genotype was compared pairwise with the wild type (Col-0) for each specific treatment condition. (E) Main molecular players in the ufmylation pathway. UFSP2: UFM1-specific protease two that matures UFM1, exposing the terminal glycine residue. UBA5: the E1 activating enzyme, UFC1: E2 conjugating enzyme, UFL1: E3 ligase (F) Ufmylation pathway mutants are sensitive to ER stress triggered by tunicamycin. Root length quantification of 7-day-old seedlings grown on half strength MS media without sucrose treated with 100 ng/mL tunicamycin (Tm). *Left panel*, Root length quantification of 7-day-old seedlings. n ≈ 100 seedlings per genotype and treatment. *Right Panel*, Representative images of 7-day-old seedlings grown in described conditions. Scale bars, 5 mm. To the left are non-treated seedlings, to the right are seedlings grown at 100 ng/mL Tm. (G) AtC53$^{sAIM}$ mutant does not complement tunicamycin sensitivity phenotype. Root length quantification of indicated 7-day-old seedlings grown on half strength MS media without sucrose in control conditions (Ctrl) or treated with 100 ng/mL tunicamycin (Tm). T1 transgenic lines were used. n = 12 seedlings per genotype and treatment. Data represent the median with its interquartile range. Root length of each genotype was compared pairwise with the wild type (Col-0) for each treatment condition. Significant differences compared to control treatment (Ctrl) are indicated with * when p value $\leq$ 0.05, ** when p value $\leq$ 0.01, and *** when p value $\leq$ 0.001.

The online version of this article includes the following figure supplement(s) for figure 10:

**Figure supplement 1.** C53 and the UFMylation machinery are essential for ER stress tolerance.

## Autophagy flux assay in *Arabidopsis thaliana*

20–30 seedlings for western blot or 0.5–1 g seedlings for immunoprecipitation and mass spectrometry were grown in liquid culture for 5 days under continuous light with shaking at 80 rpm. Media was supplemented with different drugs (3 µM Torin, 10 µg/ml Tunicamycin or other drugs dissolved in DMSO) as indicated. 1 µM of concanamycin was added, if indicated in figures, to track the contribution of vacuolar degradation. For nutrient starvation, seedlings were transferred to phosphate, nitrogen- or sucrose-depleted media (–C, –P, -N). The plants were kept in the dark to reduce sucrose production by photosynthesis or to provide drug stability. Pure DMSO was added to control samples. For analyzing total protein degradation such as TMT, seedlings were flash frozen in liquid nitrogen after 24 hr treatment. For interaction analysis such as Co-immunoprecipitation, seedling treatment was stopped after 8 hr of treatment.

Samples were homogenized in a bead mill (RetschMM300, Haan, Germany; 30 Hz, 90 s) at 4°C with zirconium oxide grinding beads or ground by mortar and pestle for bigger sample volumes. For western blotting, SDS loading buffer was added and the sample boiled at 95°C for 10 min. Lysates were cleared by centrifugation at 16,000 g for 10 min and protein concentration was normalized by Amidoblack staining (Sigma). Western blotting was performed following standard protocols as described below. 5 µg of lysate was loaded per lane.

## Human cell culture conditions

HeLa-Kyoto and HEK293T cells maintained in Dulbecco's modified Eagle's Medium (DMEM) with 10% FBS, 1% L-Glutamine and 1% Penicillin/Streptomycin. Transfection was performed with Gene-Juice transfection reagent according to manufacturer's instructions. 100 µl of empty media was mixed with 3 µL of GeneJuice and after 5 min of incubation a total of 1 µg of DNA mixture per transfection was added. After 20 min of incubation, transfection mixture was added dropwise to the cells. Cells were incubated with DNA for 24 hr. DNA containing media was removed and replaced with media. Both cell lines were authenticated using STR profiling and repeatedly tested negative for mycoplasma contamination. Testing and authentication were performed using the in house core facilities.

## Lentiviral knockdown

Lentiviral transduced shRNA-mediated knockdown of *c53* in HeLa cells:

The knockdown was performed in S2 conditions. HEK293T cells were seeded 24 hr prior to transfection in DMEM without antibiotics. At 50–60% confluency, cells were transfected with 1 µg shRNA, 750 ng psPAX2 and 250 ng pMD2.G utilizing 6 µL of GeneJuice in 250 µL of empty DMEM. After 48 hr of incubation, the virus containing media was harvested and mixed 1:1 with full media. This mixture was applied to HeLa cells that were seeded 24 hr prior. Polybrene was added to a final concentration of 4 µg/ml.

After 24 hr of incubation, the medium on target cells was exchanged with full media. After 24 hr, selection with 2 µg/ml Puromycin was started. No living cells were observed in a control plate after 24 hr. After splitting cells in S2 conditions, cells were transferred into S1 conditions.

## Autophagy flux assay in human cell culture

Cells were seeded 24 hr prior to treatment. At 50–60% confluency treatments were started by replacing media containing the indicated drugs or full media (untreated). Tunicamycin was added with a final concentration of 2.5 µg/ml and Torin with a final concentration of 3 µM. The treatments were stopped after 16 hr by removing the media and washing the cells with 1xPBS. A 2 hr recovery period was started by adding either media containing 100 nM Bafilomycin A1 or full media. Cells were put on ice and lysed with 100 µL of Lysis buffer (50 mM HEPES, 150 mM NaCl, 1 mM EDTA, 1 mM EGTA, 25 mM NaF, 10 µM $ZnCl_2$, 1% Triton X-100% and 10% Glycerol) per well. After centrifugation, supernatant was mixed 1:1 with 2x Laemmli Buffer and denatured by heating to 95℃ for 5 min.

Each sample was loaded onto a 4–20% SDS-PAGE gradient gel (BioRad) and electrophoresis was run at 100V for 1.5 hr.

## Western blotting

SDS-PAGE was performed using gradient 4–20% Mini-PROTEAN TGX Precast Protein Gels (BioRad). Blotting on nitrocellulose membranes was performed using a semi-dry Turbo transfer blot system (BioRad). For images of human LC3B, a wet transfer to PVDF membranes was performed at 200 mA for 70 min. Membranes were blocked with 5% skimmed milk or BSA in TBS and 0.1% Tween 20 (TBS-T) for 1 hr at room temperature or at 4℃ overnight. This was followed by incubation with primary and subsequent secondary antibody conjugated to horseradish peroxidase. After three times 10 min washes with TBS-T, the immune-reaction was developed using ECL Super-Pico Plus (Thermo) and detected with ChemiDoc Touch Imaging System (BioRad).

## Western blot image quantification

Protein bands intensities were quantified with Image Lab 6 (BioRad). Equal rectangles were drawn around the total protein gel lane and the band of interest. The lane profile was obtained by subtracting the mean intensity of the background. The adjusted volume of the peak in the profile was taken as a measure of the band intensity. The protein band of interest was normalized for the total protein level of the whole lane. Average relative intensities and a standard error of at least three independent experiments were calculated.

## Chemicals and antibodies

To generate *At*C53 antibody, purified protein was sent to Eurogentec for immunization of rabbits via their 28 day program. The final bleed was purified on column conjugated with the purified protein.

## In vitro pulldowns

For pulldown experiments, 10 µl of glutathione magnetic agarose beads (Pierce Glutathione Magnetic Agarose Beads, Thermo Scientific) were equilibrated by washing them two times with wash buffer (100 mM Sodium Phosphate pH 7.2, 300 mM NaCl, 1 mM DTT, 0.01% (v/v) IGEPAL). Normalized *E. coli* clarified lysates or purified proteins were mixed, according to the experiment, added to the washed beads and incubated on an end-over-end rotator for 1 hr at 4℃. Beads were washed five times in 1 ml wash buffer. Bound proteins were eluted by adding 100 µl Laemmli buffer. Samples were analysed by western blotting or Coomassie staining.

## Yeast two hybrid assay (Y2H)

Yeast two hybrid assay (Y2H) was performed according to the Mathmaker GAL4 Two hybrid system (Clonetech) following the protocol from the manufacture. Different genes were fused in frame to GAL4 activation domain of the prey vector pGADT7 and GAL4 binding domain from the bait vector pGBKT7. Split-GFP was used as positive control. Combinations of pGADT7 and pGBKT7 vectors carrying the different genes were transformed in the yeast strains Y187 (MAT α) and AH109 (MAT a),

respectively. After mating between bait and prey strains, the diploid yeast was selected for growth on (SD)-Leu /- Trp, (SD)-Leu /- Trp /- His and (SD)-Leu /- Trp /- His/-Ade plates at 28°C for 2 to 4 days.

## In planta co-immunoprecipitation

0.5–1 g seedlings were grown in liquid and treated as described under section Autophagy Flux Assay. After homogenization of frozen samples by bead-mill, G-TEN buffer (10% Glycerole, 50 mM Tris/HCl pH 7.5, 1 mM EDTA, 300 mM NaCl, 1 mM DTT, 0.1% [v/v] Nonidet P-40/Igepal, Complete protease inhibitor tablet) was added, vortexed, and lysates were cleared by centrifugation at 16,000 g for 10 min at 4°C. Protein concentration was equally adjusted using Bradford protein assay (Sigma).

25 µl of RFP or GFP-Trap_A beads (Chromotek) were equilibrated and added to each lysate and incubated for 2 hr at 4°C on a turning wheel. Beads were washed three times with 1 mL G-TEN buffer.

For western Blot analysis, beads were resuspended in 30 µl SDS-loading buffer (116 mM Tris-HCl pH 6.8, 4.9% glycerol, 10 mM DTT, 8% SDS). On-bead bound proteins were eluted by boiling the beads for 10 min at 70°C and analysed by western blotting with indicated antibodies.

For mass spectrometry experiments, the beads were further washed five times with mass spectrometry compatible buffer (50 mM Tris/HCl pH 7.5, 1 mM EDTA). Buffer resuspended beads were then submitted for trypsin digestion.

## Microscopy-based protein-protein interaction assays

Bead-bound bait proteins were incubated with fluorescently labelled prey protein as described previously by *Turco et al., 2019*.

10 µl of Glutathione Sepharose 4B beads (GE Healthcare, average diameter 90 mm) were incubated for 30 min at 4°C (16 rpm horizontal rotation) with GST-tagged bait proteins (4 mg/mL for GST and GST-FIP200 CTR). The beads were washed two times in 10x bead volume of washing buffer (25 mM HEPES pH 7.5, 150 mM NaCl, 1 mM DTT). The buffer was removed, and the beads were resuspended 1:1 in washing buffer. 10 µL of a 2–5 µM dilution of fluorescently labeled binding partners (GFP, C53-GFP and GFP-p62) were added to the bead suspension and incubated for 30–60 min at room temperature before imaging with a Zeiss LSM700 confocal microscope with 20 X magnification. For quantification, the maximum gray value along the diameter of each bead (n ≥ 15) was measured.

## Mass spectrometry (TMT) and analysis

MS/MS Data analysis: Raw files were processed with Proteome Discoverer (version 2.3, Thermo Fisher Scientific, Bremen, Germany). Database searches were performed using MS Amanda (version 2.3.0.14114) (*Dorfer et al., 2014*) against the TAIR10 database (32785 sequences). The raw files were loaded as fractions into the processing workflow. Carbamidomethylation of cysteine and TMT on peptide N-termini were specified as fixed modifications, phosphorylation on serine, threonine and tyrosine, oxidation of methionine, deamidation of asparagine and glutamine, TMT on lysine, carbamylation on peptide N-termini and acetylation on protein N-termini were set as dynamic modifications. Trypsin was defined as the proteolytic enzyme, cleaving after lysine or arginine. Up to two missed cleavages were allowed. Precursor and fragment ion tolerance were set to 5 ppm and 15 ppm, respectively. Identified spectra were rescored using Percolator (*Käll et al., 2007*), and filtered to 0.5% FDR at the peptide spectrum match level. Protein grouping was performed in Proteome Discoverer applying strict parsimony principle. Proteins were subsequently filtered to a false discovery rate of 1% at protein level. Phosphorylation sites were localized using IMP-ptmRS implemented in Proteome Discoverer using a probability cut-off of >75% for unambiguous site localization.

TMT-quantification: TMT reporter ion S/N values were extracted from the most confident centroid mass within an integration tolerance of 20 ppm. PSMs with average TMT reporter S/N values below 10 as well as PSMs showing more than 50% co-isolation were removed. Protein quantification was determined based on unique peptides only. Samples were sum normalized and missing values were imputed by the 5% quantile of the reporter intensity in the respective sample. Statistical significance of differentially abundant proteins was determined using limma (*Smyth, 2004*). Gene

Ontology (*Ashburner et al., 2000*) enrichment was determined using DAVID (*Dennis et al., 2003*) (version 6.8). Cross species comparison of regulated proteins was performed by mapping proteins to ortholog clusters available in eggnog (*Huerta-Cepas et al., 2016*). Proteins containing signal peptides were predicted using SignalP 5.0 (*Almagro Armenteros et al., 2019*).

## Peptide array

High-density peptide array analysis was performed commercially by PEPperPRINT. This comprised a full substitution scan of wild-type peptide GVSEWDPILEELQEM, with exchange of all amino acid positions with 23 amino acids including citrulline (Z), methyl-alanine (O) and D-alanine (U). The analysis also included an N- and C-terminal deletion series of wild-type peptide GVSEWDPILEELQEM; an additional 32 spots of custom control peptide KPLDFDWEIVLEEQ, and acidic variants of this control peptide involving exchanges of selected amino acid positions with glutamic acid €. The resulting peptide microarrays contained 416 different linear peptides printed at least in triplicate (1412 peptide spots; wild-type peptides were printed with a higher frequency), and were framed by HA (YPYDVPDYAG, 88 spots) control peptides (See *Supplementary file 1* for the array map).

Peptide microarrays were pre-stained with rabbit anti-GST Dylight680 at a dilution of 1:2000 to investigate background interactions with the variants of wild-type peptides GVSEWDPILEELQEM and KPLDFDWEIVLEEQ that could interfere with the main assays. Subsequent incubation of other peptide microarrays with proteins GST-ATG8A and GST at a concentration of 10 µg/ml in incubation buffer was followed by staining with secondary antibody rabbit anti-GST Dylight680 and read-out at a scanning intensity of 7 (red). The control staining of the HA epitopes with control antibody mouse monoclonal anti-HA (12CA5) DyLight800 was finally done as an internal quality control to confirm the assay quality and the peptide microarray integrity. Read-out of the control staining was performed at a scanning intensity of 7/7 (red/green).

Quantification of spot intensities and peptide annotation were based on the 16-bit grey scale tiff files at a scanning intensity of 7 that exhibit a higher dynamic range than the 24-bit colorized tiff files; microarray image analysis was done with PepSlide Analyzer. A software algorithm breaks down fluorescence intensities of each spot into raw, foreground and background signal (see 'Raw Data' tabs), and calculates averaged median foreground intensities and spot-to-spot deviations of spot duplicates.

## Microscopy methods

### Preparation of *Arabidopsis thaliana* samples for confocal imaging

Four-day-old seedlings were treated as indicated under the autophagy flux assay section. Seedlings were imaged between 3 hr - 6 hr of drug incubation. Roots were placed on a microscope slide with indicated treatment buffer and closed with coverslip. Imaging was performed in the root differentiation zone where root hair growth starts.

### Preparation of human cell samples for confocal imaging

Transfected and treated cells were grown on coverslips and fixed utilizing 0.4% Paraformaldehyde solution in PBS for 30 min. Fixed cells were mounted in VectaShield mounting medium without DAPI and sealed using clear nail polish.

### Confocal imaging

Samples were imaged at an upright ZEISS LSM800 or LSM 780 confocal microscope (Zeiss) with an Apochromat 40x or 63x objective lens at 1x magnification.

Excitation/detection parameters for GFP and mCherry were 488 nm/463 nm and 510 nm and 561 nm/569 to 635 nm, respectively, and sequential scanning mode was used for colocalization of both fluorophores. Identical settings, including an optical section thickness of 2 µm per z-stack, were used during the acquisition for sample comparison, and the images processed using identical parameters. Confocal images were processed with ZEN (version 2011) and ImageJ (version 1.48 v) software.

### Image quantification

Autophagic puncta were counted using ImageJ. Several (at least five) z-stack merged images were manually background subtracted, thresholded and the same threshold value was applied to all the

images and replicates of the same experiment. The image was converted to eight-bit grayscale and then counted for ATG8 puncta either manually or by the Particle Analyzer function of ImageJ. The average number of autophagosomes per z-stack was averaged between 10 or more different roots.

Colocalization analysis was performed by calculating Pearson's correlation coefficient as previously described using ImageJ software with the plug-in JACoP (*Bolte and Cordelières, 2006*). Values near one represent almost perfect correlation, whereas values near 0 reflect no correlation. The average Pearson's correlation coefficient was determined in five or more different roots.

## Ultrastructural analyses using immunogold labeling electron microscopy

### TEM experiments using mCherry and native AtC53 antibodies

For high-pressure freezing, 5-day-old *Arabidopsis* seedling roots expressing AtC53-mCherry were cut and high-pressure frozen (EM PACT2, Leica, Germany), prior to subsequent freeze substitution in acetone containing 0.4% uranyl acetate at −85˚C in an AFS freeze-substitution unit (Leica, Wetzlar, Germany). After gradient infiltration with increasing concentration of HM20, root samples were embedded and ultraviolet polymerized for ultra-thin sectioning and imaging. TEM images were captured by an 80 kV Hitachi H-7650 transmission electron microscope (Hitachi High-Technologies Corporation, Japan) with a charge-coupled devise camera. IEM analysis were performed as previously described (*Zhuang et al., 2017*).

### TEM experiments using GFP antibodies

*Arabidopsis* roots were fixed in 2% paraformaldehyde and 0.2% glutaraldehyde (both EM-grade, EMS, USA) in 0.1 M PHEM buffer (pH 7) for 2 hr at RT, then overnight at 4˚C. The fixed roots were embedded in 12% gelatin and cut into 1 mm$^3$ blocks which were immersed in 2.3 M sucrose overnight at 4˚C. These blocks were mounted onto a Leica specimen carrier (Leica Microsystems, Austria) and frozen in liquid nitrogen. With a Leica UCT/FCS cryo-ultramicrotome (Leica Microsystems, Austria) the frozen blocks were cut into ultra-thin sections at a nominal thickness of 60 nm at −120˚C. A mixture of 2% methylcellulose (25 centipoises) and 2.3 M sucrose in a ratio of 1:1 was used as a pick-up solution. Sections were picked up onto 200 mesh Ni grids (Gilder Grids, UK) with a carbon coated formvar film (Agar Scientific, UK). Fixation, embedding and cryo-sectioning was conducted as described by *Tokuyasu, 1973*.

### Immunolabeling

Prior to immunolabeling, grids were placed on plates with solidified 2% gelatine and warmed up to 37˚C for 20 min to remove the pick-up solution. After quenching of free aldehyde-groups with glycine (0.1% for 15 min), a blocking step with 1% BSA (fraction V) in 0.1 M Sörensen phosphate buffer (pH 7.4) was performed for 40 min. The grids were incubated in primary antibody, rabbit polyclonal to GFP (ab6556, Abcam, UK), diluted 1:125 in 0.1 M Sörensen phosphate buffer over night at 4˚C, followed by a 2 hr incubation in the secondary antibody, a goat-anti-rabbit antibody coupled with 6 nm gold (GAR 6 nm, Aurion, The Netherlands), diluted 1:20 in 0.1 M Sörensen phosphate buffer, performed at RT. The sections were stained with 4% uranyl acetate (Merck, Germany) and 2% methylcellulose at a ratio of 1:9 (on ice). All labeling steps were conducted in a wet chamber. The sections were inspected using a FEI Morgagni 268D TEM (FEI, The Netherlands) operated at 80kV. Electron micrographs were acquired using an 11-megapixel Morada CCD camera from Olympus-SIS (Germany).

## Statistical analysis

Statistical analyses were performed with GraphPad Prism eight software. For all the quantifications described above, statistical analysis was performed. Statistical significance of differences between two experimental groups was assessed wherever applicable by either a two-tailed Student's t-test if the variances were not significantly different according to the F test, or using a non-parametric test (Mann-Whitney or Kruskal-Wallis with Dunn's post-hoc test for multiple comparisons) if the variances were significantly different ($p < 0.05$). Differences between two data sets were considered significant at $p < 0.05$ (*); $p < 0.01$ (**); $p < 0.001$ (***); $p < 0.0001$ (****); n.s., not significant.

## Biophysical characterization

### Protein purification

Recombinant proteins were produced using *E. coli* strain Rosetta2(DE3)pLysS grown in 2x TY media at 37°C to an A600 of 0.4–0.6 followed by induction with 300 μM IPTG and overnight incubation at 18°C. Pelleted cells were resuspended in lysis buffer (100 mM Sodium Phosphate pH 7.0, 300 mM NaCl) containing protease inhibitors (Complete, Roche) and sonicated. The clarified lysate was first purified by affinity, by using HisTrap FF (GE HealthCare) columns. The proteins were eluted with lysis buffer containing 500 mM Imidazole. The eluted fraction was buffer exchanged to 10 mM Sodium Phosphate pH 7.0, 100 mM NaCl and loaded either on Cation Exchange, Resource S, or Anion Exchange, Resource Q, chromatography columns. The proteins were eluted by NaCl gradient (50% in 20 CV). Finally, the proteins were separated by Size Exclusion Chromatography with HiLoad 16/600 Superdex 200 pg or HiLoad 16/600 Superdex 75 pg, which were previously equilibrated in 50 mM Sodium Phosphate pH 7.0, 100 mM NaCl. The proteins were concentrated using Vivaspin concentrators (3000, 5000, 10000 or 30000 MWCO). Protein concentration was calculated from the UV absorption at 280 nm by DS-11 FX+ Spectrophotometer (DeNovix).

### Surface plasmon resonance analysis

Binding of AIM *wt* (EPLDFDWEIVLEEEM) and AIM mutant (EPLDFDAEIALEEEM) peptide to GST-GABARAP and GST-ATG8A, respectively, was investigated by surface plasmon resonance analysis using a Biacore T200 instrument (GE Healthcare) operated at 25°C. In additione, AIM-dependent binding of *Hs*C53 and *At*C53 to GST-GABARAP and GST-ATG8A were studied. The running buffer used for all experiments was 50 mM sodium phosphate pH 7.0 supplemented with 100 mM NaCl, 0.05% (v/v) Tween-20% and 0.1% (w/v) bovine serum albumin.

Polyclonal anti-GST antibodies (GST Capture Kit, GE Healthcare) were amine coupled on to a Series S CM5 sensor chip (GE Healthcare) using two adjacent flow cells (i.e. the reference and active cell) according to the manufacturer's instructions.

To determine specific binding, GST-GABARAP or GST-ATG8A were captured on the active cell (concentration: 5 μg/ml; contact time: 30 s; flow rate: 10 μl/min) and GST was captured on the reference cell (concentration: 10 μg/ml; contact time: 30 s; flow rate: 10 μl/min) to perform background subtraction.

To qualitatively show whether the analytes, *Hs*C53, *Hs*C53 123A (i.e. HsC53$^{W269A, W294A, W312A}$), *At*C53 and *At*C53 1234A (i.e. *At*C53$^{W276A, W287A, Y304A, W335A}$), interact or do not interact in an AIM-dependent manner with GST-GABARAP or GST-ATG8A, the two flow cells were exposed to four sets of double consecutive injections (1st set: 10 μM analyte, running buffer; 2nd set: 10 μM analyte, 10 μM analyte; 3rd set: 10 μM analyte, 10 μM analyte + 6.4 μM AIM *wt* peptide; 4th set: 10 μM analyte, 10 μM analyte + 6.4 μM AIM mutant peptide. Contact time 1st injection: 30 s; contact time 2nd injection: 30 s; dissociation time: 60 s; flow rate: 30 μl/min).

To quantify the binding affinities of the AIM *wt* peptide to GST-ATG8 or GST-GABARAP, multi-cycle kinetic experiments with increasing concentrations of the AIM *wt* peptide (25, 50, 100, 200, 400, 800, 1600, 3200 nM and 400 nM as internal replicates) were performed (contact time: 60 s; dissociation time: 60 s; flow rate: 30 μl/min). As a negative control, the chip was exposed to 3200 nM of the AIM mutant peptide (contact time: 60 s; dissociation time: 60 s; flow rate: 30 μl/min).

To quantify the apparent binding affinity of the AIM *wt* peptide to GST-GABARAP in presence of *Hs*C53, multi-cycle kinetic experiments with increasing concentrations of the AIM peptide (0, 25, 50, 100, 200, 400, 800, 1600, 3200 nM and 400 nM as internal replicates), containing 10 μM of *Hs*C53, in running buffer (contact time: 60 s; dissociation time: 60 s; flow rate: 30 μl/min). For negative controls, the chip was exposed to 3200 nM of the AIM mutant peptide, containing 10 μM of *Hs*C53 or 10 μM of *Hs*C53 123A, and to 3200 nM of the AIM *wt* peptide, containing 10 μM of *Hs*C53 123A (contact time: 60 s; dissociation time: 60 s; flow rate: 30 μl/min).

To quantify the apparent binding affinity of the AIM *wt* peptide to GST-ATG8A in the presence of *At*C53, multi-cycle kinetic experiments with increasing concentrations of the AIM peptide (0, 50, 100, 200, 400, 800, 1600, 3200, 6400, 12800 nM and 400 nM as internal replicate) containing 10 μM of *At*C53 were performed (contact time: 60 s; dissociation time: 60 s; flow rate: 30 μl/min). As negative controls, the chip was exposed to 6400 nM of the AIM mutant peptide, containing 10 μM of

AtC53 or 10 µM of AtC53 1234A, and to 6400 nM of the AIM wt peptide, containing 10 µM of AtC53 1234A (contact time: 60 s; dissociation time: 60 s; flow rate: 30 µl/min).

After each cycle, regeneration was performed with 2 injections of 10 mM glycine-HCl pH 2.1 for 120 s at a flow rate of 10 µL/min.

The sensograms obtained were analyzed with Biacore T200 Evaluation software (version 3.1) by global fitting of the data to a 1:1 steady-state affinity model.

Molecular weights and sources of the proteins for the SPR experiments are reported in the following table:

| Sample name | Source | MW (Da) |
|---|---|---|
| AIM wt peptide | Synthetized in house | 1894.08 |
| AIM mutant peptide | | 1750.89 |
| HsC53 | Escherichia coli recombinant expression | 59191.15 |
| HsC53 123A | | 58758.66 |
| AtC53 | | 64399.57 |
| AtC53 1234A | | 63962.07 |
| GST-GABARAP | | 42458.87 |
| GST-ATG8A | | 42366.85 |
| GST | | 27898.33 |

Calculation for the apparent $K_D$ ($K_D^{'}$) of the AIM$^{wt}$ was done by using the following formula (Nelson, David L. Lehninger Principles Of Biochemistry. New York: W.H. Freeman, 2008):

$$K_D^{'} AIM^{wt} = \alpha K_D(AIM^{wt})$$

Where, $\alpha = 1 + \frac{I}{K_{i'}}$, $K_i = K_D(C53)$ and $[I] = [C53]$
Then:

$$K_D(C53) = \frac{K_D(AIM^{wt})[C53]}{K_D^{'}(AIM^{wt}) - K_D(AIM^{wt})}$$

## Isothermal titration calorimetry (ITC)

All experiments were carried out at 25°C in 50 mM sodium phosphate buffer pH 7.0, 100 mM NaCl, using the PEAQ-ITC Automated (Malvern Panalytical Ltd). For protein-protein interactions, the calorimetric cell was filled with 40 µM GABARAP or ATG8A and titrated with 250 µM HsC53 or AtC53 IDRs, respectively. A single injection of 0.4 µl of HsC53 or AtC53 IDRs (not taken into account) was followed by 18 injections of 2 µl each. Injections were made at 150 s intervals with a duration of 4 s and a stirring speed of 750 rpm. The reference power was set to 10 µcal/s, the feedback mode was set to high. For protein-peptide interactions, the calorimetric cell was filled with 40 µM GABARAP or ATG8A and titrated with 600 µM peptide from the syringe. The titrations were performed as described above. For the control experiments, equivalent volumes of the IDRs, or the peptides, were titrated to buffer, equivalent volumes of buffer were titrated to GABARAP or ATG8A and equivalent volumes of buffer were titrated to buffer, using the parameters above. The raw titration data were integrated, corrected for the controls and fitted to a one-set-of-sites binding model using the PEAQ-ITC analysis software (Version 1.22).

## Sample preparation for native MS experiments

Proteins were buffer exchanged into ammonium acetate using BioRad Micro Bio-Spin 6 Columns and concentrations were measured using DS-11 FX+ Spectrophotometer (DeNovix).

## Mass spectrometry measurements

Native mass spectrometry experiments were carried out on a Synapt G2Si instrument (Waters, Manchester, UK) with a nanoelectrospray ionisation source. Mass calibration was performed by a separate infusion of NaI cluster ions. Solutions were ionised through a positive potential applied to

metal-coated borosilicate capillaries (Thermo Scientific). The following instrument parameters were used; capillary voltage 1.3 kV, sample cone voltage 40 V, extractor source offset 30 V, source temperature 40°C, trap gas 3 mL/min. A higher capillary voltage (1.9 kV) was required for ionization of the 1:2 AtC53-AtG8A complex. Data were processed using Masslynx V4.1 and spectra were plotted using R. Peaks were matched to protein complexes by comparing measured m/z values with expected m/z values calculated from the mass of individual proteins which are given in table below.

| Protein | Expected mass from sequence/Da | Measured Mass/Da |
| --- | --- | --- |
| AtC53 | 64 399.6 | 64 401.3 |
| AtC53-1234A | 63 962.1 | 63 976.7 |
| HsC53 | 59 191.1 | 59 193.0 |
| AtG8A | 15 965.3 | 15 964.0 |
| GABARAP | 15 968.3 | 15 968.2 |
| AtC53-IDR | 9050.5 | 9050.5 |
| HsC53-IDR | 6060.2 | 6059.5 |

## Circular dichroism spectroscopy

CD spectroscopy experiments were performed using a Chirascan-Plus CD spectrophotometer (Applied Photophysics). Purified proteins in 50mM sodium phosphate pH 7.0, 100mM NaCl were diluted to approximately 0.2 mg/ml and spin-filtered with an 0.1μm filter. CD measurements were carried out in a quartz glass cuvette with 0.5 mm path length. To obtain overall CD spectra, wavelength scans between 180 nm and 260 nm were collected at 25°C using a 1.0 nm bandwidth, 0.5 nm step size, and time per point of 0.5 s. Both CD and absorbance data were collected at the same time over three accumulations and averaged. CD data at wavelengths where the absorptivity was above 2.5 are not shown (data below 194nm). The raw data in millidegree units were corrected for background and drift ($\Theta_{dcorr}$). Subsequently, the differential molar extinction coefficient per peptide bond ($\Delta\varepsilon$) was calculated, taking into account the absorptivity measured at 205 nm ($A_{205}$) and the calculated protein extinction coefficient at 205 nm ($\varepsilon_{205}$) using the equation

$$\varepsilon = \frac{\Theta_{dcorr} \cdot \varepsilon_{205}}{10 \cdot A_{205} \cdot (N-1) \cdot 3298}$$

## Acknowledgements

We thank R Strasser, M Schuldiner, R Kopito, J Christianson, K Mukhtar, S Howell, D Hofius, R Vierstra, Y Ye, W Yarbrough and T Ueda for sharing plasmids or plant lines. We acknowledge the Vienna BioCenter Core Facilities GmbH (VBCF) facilities Plant Sciences (J Jez), Electron Microscopy (T Heuser, S Jakob, M Brandstetter), and Protein Technologies (P Stolt-Bergner, A Sedivy, J Neuhold, A Lehner) as well as the GMI/IMBA/IMP Protein Chemistry Core facility (M Madalinski, E Roitinger, K Stejskal, R Imre) and A Schleiffer for their help with the experiments. The SPR and ITC equipment were kindly provided by the EQ-BOKU VIBT GmbH and the BOKU Core Facility Biomolecular and Cellular Analysis. We thank members of the *Vienna Biocenter Ubiquitin Club* for fruitful discussions. This work has been funded by the Vienna Science and Technology Fund (WWTF) through project LS17-047 (YD, TC), the Austrian Science Fund (FWF): P32355 (YD), P30401-B21 (SM), I3033-B22 (AD), Unidocs Fellowship (AS), ERC grant No.646653 (SM), and the Austrian Academy of Sciences. MM is financially supported by The Financial Supports for Young Scientists (WULS-SGGW) International Research Scholarship Fund No. BWM 315/2018. We thank Claudine Kraft, Elif Karagoz, James Watson and Youssef Belkhadir for critical evaluation of the manuscript. We also acknowledge Sabbi Lall, Life Science Editors for editing assistance.

# Additional information

## Competing interests

Sascha Martens: is member of the scientific advisory board of Casma Therapeutics. The other authors declare that no competing interests exist.

## Funding

| Funder | Grant reference number | Author |
| --- | --- | --- |
| Vienna Science and Technology Fund | LS17-047 | Madlen Stephani<br>Lorenzo Picchianti<br>Tim Clausen<br>Yasin Dagdas |
| Austrian Science Fund | P32355 | Yasin Dagdas |
| Austrian Science Fund | P30401-B21 | Sascha Martens |
| Austrian Science Fund | I3033-B22 | Armin Djamei |
| Austrian Science Fund | Unidocs fellowship | Adriana Savova |
| Austrian Academy of Sciences | | Alexander Gajic<br>Emilio Skarwan<br>Victor Sanchez de Medina Hernandez<br>Azadeh Mohseni<br>Marion Clavel<br>Christian Loefke<br>Alibek Abdrakhmanov<br>Yasin Dagdas |
| Horizon 2020 Framework Programme | No.646653 | Sascha Martens |
| The Financial Supports for Young Scientists (WULS-SGGW) International Research Scholarship Fund | BWM 315/2018 | Mateusz Matuszkiewicz |

The funders had no role in study design, data collection and interpretation, or the decision to submit the work for publication.

## Author contributions

Madlen Stephani, Lorenzo Picchianti, Conceptualization, Resources, Data curation, Formal analysis, Validation, Investigation, Visualization, Methodology, Writing - original draft, Writing - review and editing; Alexander Gajic, Investigation, Visualization, Methodology, Writing - original draft, Writing - review and editing; Rebecca Beveridge, Conceptualization, Formal analysis, Investigation, Visualization, Writing - original draft, Writing - review and editing; Emilio Skarwan, Formal analysis, Visualization, Writing - original draft; Victor Sanchez de Medina Hernandez, Khong-Sam Chia, Investigation, Writing - original draft; Azadeh Mohseni, Data curation, Investigation, Visualization, Writing - original draft; Marion Clavel, Yonglun Zeng, Christin Naumann, Mateusz Matuszkiewicz, Alibek Abdrakhmanov, Investigation, Visualization, Writing - original draft; Eleonora Turco, Baiying Li, Investigation, Visualization; Christian Loefke, Adriana Savova, Resources; Gerhard Dürnberger, Data curation, Formal analysis, Investigation, Visualization; Michael Schutzbier, Data curation, Investigation, Methodology; Hsiao Tieh Chen, Investigation; Armin Djamei, Steffen Abel, Liwen Jiang, Fumiyo Ikeda, Supervision; Irene Schaffner, Supervision, Investigation, Methodology, Writing - review and editing; Karl Mechtler, Supervision, Methodology; Sascha Martens, Supervision, Funding acquisition, Writing - review and editing; Tim Clausen, Conceptualization, Supervision, Funding acquisition, Writing - original draft, Writing - review and editing; Yasin Dagdas, Conceptualization, Formal analysis, Supervision, Funding acquisition, Investigation, Writing - original draft, Project administration, Writing - review and editing

## Author ORCIDs
Madlen Stephani (iD) https://orcid.org/0000-0001-6614-833X
Armin Djamei (iD) https://orcid.org/0000-0002-8087-9566
Fumiyo Ikeda (iD) http://orcid.org/0000-0003-0407-2768
Sascha Martens (iD) http://orcid.org/0000-0003-3786-8199
Tim Clausen (iD) https://orcid.org/0000-0003-1582-6924
Yasin Dagdas (iD) https://orcid.org/0000-0002-9502-355X

## Decision letter and Author response
Decision letter https://doi.org/10.7554/eLife.58396.sa1
Author response https://doi.org/10.7554/eLife.58396.sa2

# Additional files
## Supplementary files
• Supplementary file 1. Peptide array map.

• Supplementary file 2. IP/MS-analyses of mCherry, mCherry-ATG8A and mCherry-ATG8E.

• Supplementary file 3. Quantitative proteomics analyses of AtC53 mediated degradation (TMT).

• Supplementary file 4. Enriched GO categorization of quantitative proteomics analyses of AtC53 mediated degradation (TMT).

• Supplementary file 5. IP/MS analyses of YFP, C53-GFP, UFL1-GFP and DDRGK1-GFP.

• Supplementary file 6. Summary of thermodynamic parameters of the interactions studied in this paper.

• Transparent reporting form

## Data availability
All the raw data associated with the figures are uploaded to Dryad and accessible here doi:10.5061/dryad.wm37pvmkb. The mass spectrometry proteomics data have been deposited to the ProteomeXchange Consortium via the PRIDE partner repository with the dataset identifier PXD019988.

The following datasets were generated:

| Author(s) | Year | Dataset title | Dataset URL | Database and Identifier |
|---|---|---|---|---|
| Stephani M, Dürnberger G, Schutzbier M, Imre R, Mechtler K, Dagdas Y | 2020 | Mass Spectrometry Proteomics Data (Quantitiative Proteomics/TMT, IP-MS) | http://proteomecentral.proteomexchange.org/cgi/GetDataset?ID=PXD019988 | ProteomeXchange, PXD019988 |
| Stephani M, Picchianti L, Gajic A, Beveridge R, Skarwan E, Sanchez V, de Medina H, Mohseni A, Zeng Y, Naumann C, Matuszkiewicz M, Turco E, Li B, Dürnberger G, Schutzbier M, Chen HT, Abdrakhmanov A, Chia KS, Schaffner I, Dagdas Y | 2020 | Raw data corresponding to all experiments presented in the research article | https://doi.org/10.5061/dryad.wm37pvmkb | Dryad Digital Repository, 10.5061/dryad.wm37pvmkb |

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

# Appendix 1

## Key resources table

**Appendix 1—key resources table**

| Reagent type (species) or resource | Designation | Source or reference | Identifiers | Additional information |
|---|---|---|---|---|
| Genetic reagent (*Arabidopsis thaliana*) | Col-0 | | | |
| Genetic reagent (*Marchantia Polymorpha*) | Tak-1 | | | |
| Cell line (*Homo sapiens*) | HeLa-Kyoto | Fumiyo Ikeda | | See Affiliations |
| Cell line (*Homo sapiens*) | HEK293T | Fumiyo Ikeda | | See Affiliations |
| Genetic reagent (*Arabidopsis thaliana*) | c53 | this study | At5g06830 | See Methods, CRISPR/Cas9 construct design. Available on request to the corresponding authors. |
| Genetic reagent (*Arabidopsis thaliana*) | ddrgk1 | this study | At4g27120 | See Methods, CRISPR/Cas9 construct design. Available on request to the corresponding authors. |
| Genetic reagent (*Arabidopsis thaliana*) | ufm1 | this study | At1g77710 | See Methods, CRISPR/Cas9 construct design. Available on request to the corresponding authors. |
| Genetic reagent (*Arabidopsis thaliana*) | atg2 | Morten Peterson Wang et al. Plant Journal (2011) | At3g19190 | EMS-mutant (Gln803stop) |
| Genetic reagent (*Arabidopsis thaliana*) | atg5 | NASC (N39993) Scholl et al. Plant Phys. (2000) | At5g17290 | SAIL_129B07 |
| Genetic reagent (*Arabidopsis thaliana*) | ufl1 | NASC (N685434) et al. Plant Phys. (2000) | At3g46220 | SALK_022517C |
| Genetic reagent (*Arabidopsis thaliana*) | uba5 | NASC (N634012) Scholl et al. Plant Phys. (2000) | At1g05350 | SALK_134012 |
| Genetic reagent (*Arabidopsis thaliana*) | ufsp2 | NASC (N826004) Scholl et al. Plant Phys. (2000) | At3g48380 | SAIL_607_G10 |
| Genetic reagent (*Arabidopsis thaliana*) | ufc1 | NASC (N678973) Scholl et al. Plant Phys. (2000) | At1g27530 | SALK_112532 |

*Continued on next page*

*Appendix 1—key resources table continued*

| Reagent type (species) or resource | Designation | Source or reference | Identifiers | Additional information |
|---|---|---|---|---|
| Genetic reagent (*Arabidopsis thaliana*) | *ire1a/b* | Karolina Pajerowska-Mukhtar McCormack et al. Front. in plant sci. (2015) | At2G17520/ At5G24360 | SALK_018112/SAIL_238_F07 |
| Genetic reagent (*Arabidopsis thaliana*) | *bzip 17/28* | Kazuo Shinozaki Kim et al. Plant Phys. (2018) | At2g40950/ At3 g10800 | SALK_104326/SALK_132285 |
| Genetic reagent (*Arabidopsis thaliana*) | *bzip28/60* | Kazuo Shinozaki Kim et al. Plant Phys. (2018) | At3g10800/ At1 g42990 | SALK_132285/SALK_050203 |
| Genetic reagent (*Arabidopsis thaliana*) | pUbi::mCherry-ATG8A | This study | | See Methods, Plant materials and Growth conditions. Available on request to the corresponding authors. |
| Genetic reagent (*Arabidopsis thaliana*) | pUbi::mCherry-ATG8E | Liwen Jiang Hu et al. J. Integr. Plant Biol. (2020) | | |
| Genetic reagent (*Arabidopsis thaliana*) | pUbi::mCherry-ATG8E x *atg5* | this study | | See Methods, Plant materials and Growth conditions. Available on request to the corresponding authors. |
| Genetic reagent (*Arabidopsis thaliana*) | pUbi::GFP-ATG8A | this study | | See Methods, Plant materials and Growth conditions. Available on request to the corresponding authors. |
| Genetic reagent (*Arabidopsis thaliana*) | pUbi::GFP-ATG8B | this study | | See Methods, Plant materials and Growth conditions. Available on request to the corresponding authors. |
| Genetic reagent (*Arabidopsis thaliana*) | pUbi::GFP-ATG8C | this study | | See Methods, Plant materials and Growth conditions. Available on request to the corresponding authors. |
| Genetic reagent (*Arabidopsis thaliana*) | pUbi::GFP-ATG8D | this study | | See Methods, Plant materials and Growth conditions. Available on request to the corresponding authors. |
| Genetic reagent (*Arabidopsis thaliana*) | pUbi::GFP-ATG8E | this study | | See Methods, Plant materials and Growth conditions. Available on request to the corresponding authors. |
| Genetic reagent (*Arabidopsis thaliana*) | pUbi::GFP-ATG8F | this study | | See Methods, Plant materials and Growth conditions. Available on request to the corresponding authors. |
| Genetic reagent (*Arabidopsis thaliana*) | pUbi::GFP-ATG8G | this study | | See Methods, Plant materials and Growth conditions. Available on request to the corresponding authors. |

*Continued on next page*

*Appendix 1—key resources table continued*

| Reagent type (species) or resource | Designation | Source or reference | Identifiers | Additional information |
|---|---|---|---|---|
| Genetic reagent (*Arabidopsis thaliana*) | pUbi::GFP-ATG8H | this study | | See Methods, Plant materials and Growth conditions. Available on request to the corresponding authors. |
| Genetic reagent (*Arabidopsis thaliana*) | pUbi::GFP-ATG8I | this study | | See Methods, Plant materials and Growth conditions. Available on request to the corresponding authors. |
| Genetic reagent (*Arabidopsis thaliana*) | pUbi::C53-mCherry | this study | | See Methods, Plant materials and Growth conditions. Available on request to the corresponding authors. |
| Genetic reagent (*Arabidopsis thaliana*) | pUbi::C53-mCherry x *atg2* | this study | | See Methods, Plant materials and Growth conditions. Available on request to the corresponding authors. |
| Genetic reagent (*Arabidopsis thaliana*) | pUbi::C53-mCherry x *atg5* | this study | | See Methods, Plant materials and Growth conditions. Available on request to the corresponding authors. |
| Genetic reagent (*Arabidopsis thaliana*) | pUbi::C53-mCherry x *ufl1* | this study | | See Methods, Plant materials and Growth conditions. Available on request to the corresponding authors. |
| Genetic reagent (*Arabidopsis thaliana*) | pUbi::C53-mCherry x *ddrgk1* | this study | | See Methods, Plant materials and Growth conditions. Available on request to the corresponding authors. |
| Genetic reagent (*Arabidopsis thaliana*) | pUbi::C53-mCherry x *ire1a/b* | this study | | See Methods, Plant materials and Growth conditions. Available on request to the corresponding authors. |
| Genetic reagent (*Arabidopsis thaliana*) | pUbi::C53-mCherry x *bzip28/60* | this study | | See Methods, Plant materials and Growth conditions. Available on request to the corresponding authors. |
| Genetic reagent (*Arabidopsis thaliana*) | pUbi::C53-mCherry x pUbi::GFP-ATG8A | this study | | See Methods, Plant materials and Growth conditions. Available on request to the corresponding authors. |
| Genetic reagent (*Arabidopsis thaliana*) | pUbi::C53-mCherry x wave-YFP | this study | | See Methods, Plant materials and Growth conditions. Available on request to the corresponding authors. |
| Genetic reagent (*Arabidopsis thaliana*) | pUbi::C53-mCherry x p35S::GFP-HDEL | this study | | See Methods, Plant materials and Growth conditions. Available on request to the corresponding authors. |
| Genetic reagent (*Arabidopsis thaliana*) | pUbi::C53-mCherry x p35S::GFP-ATG11 | this study | | See Methods, Plant materials and Growth conditions. Available on request to the corresponding authors. |
| Genetic reagent (*Arabidopsis thaliana*) | pUbi::C53-mCherry x p35S::GFP-ATG11 | this study | | See Methods, Plant materials and Growth conditions. Available on request to the corresponding authors. |

*Appendix 1—key resources table continued*

| Reagent type (species) or resource | Designation | Source or reference | Identifiers | Additional information |
|---|---|---|---|---|
| Genetic reagent (*Arabidopsis thaliana*) | pUbi::C53-mCherry x pUbi::UFL1-GFP | this study | | See Methods, Plant materials and Growth conditions. Available on request to the corresponding authors. |
| Genetic reagent (*Arabidopsis thaliana*) | pUbi::C53-mCherry x pUbi::DDRGK1-GFP | this study | | See Methods, Plant materials and Growth conditions. Available on request to the corresponding authors. |
| Genetic reagent (*Arabidopsis thaliana*) | pUbi::C53-mCherry x GCSI-SUBEX-C57Y-GFP | this study/Richard Strasser *Shin et al., 2018* | | See Methods, Plant materials and Growth conditions. Available on request to the corresponding authors. |
| Genetic reagent (*Arabidopsis thaliana*) | pUbi::C53-mCherry x MNS1-SUBEX-GFP | this study/Richard Strasser *Shin et al., 2018* | | See Methods, Plant materials and Growth conditions. Available on request to the corresponding authors. |
| Genetic reagent (*Arabidopsis thaliana*) | pUbi::C53-GFP | this study | | See Methods, Plant materials and Growth conditions. Available on request to the corresponding authors. |
| Genetic reagent (*Arabidopsis thaliana*) | pUbi::C53-GFP x *c53* | this study | | See Methods, Plant materials and Growth conditions. Available on request to the corresponding authors. |
| Genetic reagent (*Arabidopsis thaliana*) | pUbi::C53$^{sAIM(W276A, W287A,Y304A,W335A)}$-GFP x *c53* | this study | | See Methods, Plant materials and Growth conditions. Available on request to the corresponding authors. |
| Genetic reagent (*Arabidopsis thaliana*) | pUbi::C53-GFP x SP-mRFP-SUBEX-C57Y-EMP12 | this study/Richard Strasser *Shin et al., 2018* | | See Methods, Plant materials and Growth conditions. Available on request to the corresponding authors. |
| Genetic reagent (*Arabidopsis thaliana*) | pC53::C53-GFP | this study | | See Methods, Plant materials and Growth conditions. Available on request to the corresponding authors. |
| Genetic reagent (*Arabidopsis thaliana*) | pUbi::UFL1-GFP | this study | | See Methods, Plant materials and Growth conditions. Available on request to the corresponding authors. |
| Genetic reagent (*Arabidopsis thaliana*) | pUbi::DDRGK1-GFP | this study | | See Methods, Plant materials and Growth conditions. Available on request to the corresponding authors. |
| Genetic reagent (*Arabidopsis thaliana*) | pUbi::DDRGK1-GFPx pUbi::mCherryATG8A | this study | | See Methods, Plant materials and Growth conditions. Available on request to the corresponding authors. |
| Genetic reagent (*Arabidopsis thaliana*) | pUbi::DDRGK1-GFPx pUbi::mCherryATG8A x *c53* | this study | | See Methods, Plant materials and Growth conditions. Available on request to the corresponding authors. |
| Genetic reagent (*Arabidopsis thaliana*) | pUbi::DDRGK1-GFP x *atg5* | this study | | See Methods, Plant materials and Growth conditions. Available on request to the corresponding authors. |

*Appendix 1—key resources table continued*

| Reagent type (species) or resource | Designation | Source or reference | Identifiers | Additional information |
|---|---|---|---|---|
| Genetic reagent (*Arabidopsis thaliana*) | pUbi::DDRGK1-GFP x c53 | this study | | See Methods, Plant materials and Growth conditions. Available on request to the corresponding authors. |
| Genetic reagent (*Arabidopsis thaliana*) | pUbi::IRE1B-YFP x pRPS5a::C53-tagRFP | this study | | See Methods, Plant materials and Growth conditions. Available on request to the corresponding authors. |
| Genetic reagent (*Arabidopsis thaliana*) | p35S::GFP-HDEL (ER-gk) | NASC (N16251) | | |
| Genetic reagent (*Arabidopsis thaliana*) | wave-YFP (pNIGEL07 pUbi::myc-YFP) | Niko Geldner Geldner et al. The Plant Journal (2009) | | |
| Genetic reagent (*Arabidopsis thaliana*) | Wave-mCherry (pNIGEL17 pUbi::mCherry) | Niko Geldner Geldner et al. The Plant Journal (2009) | | |
| Sequence-based reagent | AtC53_BsF | this study | | ATATATGGTCTCGATTGATATCACCTTCTCTCGTCTGTT |
| Sequence-based reagent | AtC53_F0 | this study | | TGATATCACCTTCTCTCGTCTGTTTTAGAGCTAGAAATAGC |
| Sequence-based reagent | AtC53_R0 | this study | | AACCAAGGCCTTGGCTTTCTTCCAATCTCTTAGTCGACTCTAC |
| Sequence-based reagent | AtC53_BsR | this study | | ATTATTGGTCTCGAAACCAAGGCCTTGGCTTTCTTCCAA |
| Sequence-based reagent | AtDDRGK1_BsF | this study | | ATATATGGTCTCGATTGAGAGATGCTAGATCACGGGGTT |
| Sequence-based reagent | AtDDRGK1_F0 | this study | | TGAGAGATGCTAGATCACGGGGTTTTAGAGCTAGAAATAGC |
| Sequence-based reagent | AtDDRGK1_BsR | this study | | AACTGCACTTCCTCTGTAGTACCAATCTCTTAGTCGACTCTAC |
| Sequence-based reagent | AtDDRGK1_R0 | this study | | ATTATTGGTCTCGAAACTGCACTTCCTCTGTAGTACCAA |
| Sequence-based reagent | AtUFM1_BsF | this study | | ATATATGGTCTCGATTGGAGGAGATTCAGATTAGCA GTT |
| Sequence-based reagent | AtUFM1_F0 | this study | | TGGAGGAGATTCAGATTAGCA GTTTTAGAGCTAGAAATAGC |
| Sequence-based reagent | AtUFM1_R0 | this study | | AACGAAGGAGCTCCGTTCACGGCAATCTCTTAGTCGACTCTAC |

*Continued on next page*

*Appendix 1—key resources table continued*

| Reagent type (species) or resource | Designation | Source or reference | Identifiers | Additional information |
|---|---|---|---|---|
| Sequence-based reagent | *At*UFM1_BsR | this study | | ATTATTGGTCTCGAAACGAAG GAGCTCCGTTCACGGCAA |
| Sequence-based reagent | *Mp*C53- sgRNA1- FWD | this study | | CTCGTCAATCGGAAGAGACAGAGC |
| Sequence-based reagent | *Mp*C53- sgRNA1-REV | this study | | AAACGCTCTGTCTCTTCCGATTGA |
| Sequence-based reagent | *Mp*C53- sgRNA2- FWD | this study | | CTCGAAAGTTCTGCCCTGATGT |
| Sequence-based reagent | *Mp*C53- sgRNA2-REV | this study | | AAACACATCAGGGCAGAACTTT |
| Sequence-based reagent | *Mp*IRE1- sgRNA1-FWD | this study | | CTCGTACGTTAAAGGCGAATATGG |
| Sequence-based reagent | *Mp*IRE1- sgRNA1-REV | this study | | AAACCCATATTCGCCTTTAACGTA |
| Sequence-based reagent | *Mp*IRE1- sgRNA2-FWD | this study | | CTCGCATCAAAGGACCACCAGGGC |
| Sequence-based reagent | *Mp*IRE1- sgRNA2-REV | this study | | AAACGCCCTGGTGGTCCTTTGATG |
| Antibody | Anti-Rabbit IgG HRP-Conjugate (goat polyclonal) | Biorad | 1706515 | 1:10000 |
| Antibody | Anti-Mouse IgG-HRP Conjugate (goat polyclonal) | Biorad | 1706516 | 1:10000 |
| Antibody | mCherry (rabbit polyclonal) | Abcam | ab167453 | 1:5000 |
| Antibody | HIS6 (mouse monoclonal) | Sigma Aldrich | H1029 | 1:5000 |
| Antibody | GST HRP Conjugate (goat polyclonal) | GE Healthcare | RPN1236 | 1:1000 |
| Antibody | GFP (rabbit polyclonal) | Invitrogen | A11122 | 1:3000 |
| Antibody | GFP (mouse monoclonal) | Roche | 11814460001 | 1:3000 |
| Antibody | MBP (mouse monoclonal) | Sigma Aldrich | M1321-200UL | 1:3000 |
| Antibody | *Hs*C53 (mouse monoclonal) | SCBT | sc271671 | 1:1000 |
| Antibody | LC3B (mouse monoclonal) | nanoTools | 0260–100/ LC3-2G6 | 1:100 |
| Antibody | BIP3 (rabbit polyclonal) | CST | 3177 | 1:1000 |
| Antibody | Vinculin (mouse monoclonal) | Sigma Aldrich | V9131 | 1:1000 |

*Continued on next page*

*Appendix 1—key resources table continued*

| Reagent type (species) or resource | Designation | Source or reference | Identifiers | Additional information |
|---|---|---|---|---|
| Antibody | *Hs*UFM1 (rabbit monoclonal) | Abcam | ab108062 | 1:2000 |
| Antibody | ATG8A (rabbit polyclonal) | Agrisera | AS14 2811 | 1:1000 |
| Antibody | *At*C53 (rabbit polyclonal) | this study | - | 1:5000 See Methods, Chemical and Antibodies. |
| Antibody | 60S (L13) (rabbit polyclonal) | Agrisera | AS13 2650 | 1:1000 |
| Antibody | 40S (RPS14) (rabbit polyclonal) | Agrisera | AS12 2111 | 1:1000 |
| Antibody | SMT1 (rabbit polyclonal) | Agrisera | AS07 266 | 1:500 |
| Antibody | CNX1/2 (rabbit polyclonal) | Agrisera | AS12 2365 | 1:3000 |
| Antibody | BIP1/2/3 (rabbit polyclonal) | Agrisera | AS09 481 | 1:3000 |
| Recombinant DNA reagent | MBP-AtC53 | This study | | See Methods, Cloning procedures. Available on request to the corresponding authors. |
| Recombinant DNA reagent | GST-ATG8A | This study | | See Methods, Cloning procedures. Available on request to the corresponding authors. |
| Recombinant DNA reagent | GST-ATG8A$^{LDS(YL50AA)}$ | This study | | See Methods, Cloning procedures. Available on request to the corresponding authors. |
| Recombinant DNA reagent | GST-ATG8A$^{UDS}_{(IFV77AAA)}$ | This study | | See Methods, Cloning procedures. Available on request to the corresponding authors. |
| Recombinant DNA reagent | GST-ATG8B | This study | | See Methods, Cloning procedures. Available on request to the corresponding authors. |
| Recombinant DNA reagent | GST-ATG8C | This study | | See Methods, Cloning procedures. Available on request to the corresponding authors. |
| Recombinant DNA reagent | GST-ATG8D | This study | | See Methods, Cloning procedures. Available on request to the corresponding authors. |
| Recombinant DNA reagent | GST-ATG8E | This study | | See Methods, Cloning procedures. Available on request to the corresponding authors. |
| Recombinant DNA reagent | GST-ATG8F | This study | | See Methods, Cloning procedures. Available on request to the corresponding authors. |
| Recombinant DNA reagent | GST-ATG8G | This study | | See Methods, Cloning procedures. Available on request to the corresponding authors. |
| Recombinant DNA reagent | GST-ATG8H | This study | | See Methods, Cloning procedures. Available on request to the corresponding authors. |
| Recombinant DNA reagent | GST-ATG8I | This study | | See Methods, Cloning procedures. Available on request to the corresponding authors. |

*Continued on next page*

*Appendix 1—key resources table continued*

| Reagent type (species) or resource | Designation | Source or reference | Identifiers | Additional information |
|---|---|---|---|---|
| Recombinant DNA reagent | GST-GABARAP | This study | | See Methods, Cloning procedures. Available on request to the corresponding authors. |
| Recombinant DNA reagent | GST-GABARAPL1 | This study | | See Methods, Cloning procedures. Available on request to the corresponding authors. |
| Recombinant DNA reagent | GST-GABARAPL2 | This study | | See Methods, Cloning procedures. Available on request to the corresponding authors. |
| Recombinant DNA reagent | GST-LC3A | This study | | See Methods, Cloning procedures. Available on request to the corresponding authors. |
| Recombinant DNA reagent | GST-LC3B | This study | | See Methods, Cloning procedures. Available on request to the corresponding authors. |
| Recombinant DNA reagent | GST-LC3C | This study | | See Methods, Cloning procedures. Available on request to the corresponding authors. |
| Recombinant DNA reagent | GST-GABARAP$^{LDS}$ $^{(YL49AA)}$ | This study | | See Methods, Cloning procedures. Available on request to the corresponding authors. |
| Recombinant DNA reagent | GST-GABARAP$^{(P52A)}$ | This study | | See Methods, Cloning procedures. Available on request to the corresponding authors. |
| Recombinant DNA reagent | GST-GABARAP$^{(R67A)}$ | This study | | See Methods, Cloning procedures. Available on request to the corresponding authors. |
| Recombinant DNA reagent | GST-GABARAP$^{(P52A, R67A)}$ | This study | | See Methods, Cloning procedures. Available on request to the corresponding authors. |
| Recombinant DNA reagent | GST-GABARAP$^{(KK64AA)}$ | This study | | See Methods, Cloning procedures. Available on request to the corresponding authors. |
| Recombinant DNA reagent | GST-MpATG8A | This study | | See Methods, Cloning procedures. Available on request to the corresponding authors. |
| Recombinant DNA reagent | GST-MpATG8A$^{LDS(YL50AA)}$ | This study | | See Methods, Cloning procedures. Available on request to the corresponding authors. |
| Recombinant DNA reagent | GST-MpATG8B | This study | | See Methods, Cloning procedures. Available on request to the corresponding authors. |
| Recombinant DNA reagent | GST-MpATG8B$^{LDS}$ $^{(YL50AA)}$ | This study | | See Methods, Cloning procedures. Available on request to the corresponding authors. |
| Recombinant DNA reagent | MBP-MpC53 | This study | | See Methods, Cloning procedures. Available on request to the corresponding authors. |
| Recombinant DNA reagent | MBP-AtC53 | This study | | See Methods, Cloning procedures. Available on request to the corresponding authors. |
| Recombinant DNA reagent | MBP-AtC53$^{N-IDR(1-372)}$ | This study | | See Methods, Cloning procedures. Available on request to the corresponding authors. |

*Continued on next page*

*Appendix 1—key resources table continued*

| Reagent type (species) or resource | Designation | Source or reference | Identifiers | Additional information |
|---|---|---|---|---|
| Recombinant DNA reagent | MBP-AtC53$^{C\text{-}IDR(239\text{-}549)}$ | This study | | See Methods, Cloning procedures. Available on request to the corresponding authors. |
| Recombinant DNA reagent | MBP-AtC53$^{IDR(239\text{-}372)}$ | This study | | See Methods, Cloning procedures. Available on request to the corresponding authors. |
| Recombinant DNA reagent | MBP-AtC53$^{N\text{-}C(1\text{-}239, (KGSGSTSGSG)2,373\text{-}549)}$ | This study | | See Methods, Cloning procedures. Available on request to the corresponding authors. |
| Recombinant DNA reagent | MBP-HsC53 | This study | | See Methods, Cloning procedures. Available on request to the corresponding authors. |
| Recombinant DNA reagent | MBP-HsC53$^{N\text{-}IDR(1\text{-}316)}$ | This study | | See Methods, Cloning procedures. Available on request to the corresponding authors. |
| Recombinant DNA reagent | MBP-HsC53$^{C\text{-}IDR(263\text{-}506)}$ | This study | | See Methods, Cloning procedures. Available on request to the corresponding authors. |
| Recombinant DNA reagent | MBP-HsC53$^{IDR(263\text{-}316)}$ | This study | | See Methods, Cloning procedures. Available on request to the corresponding authors. |
| Recombinant DNA reagent | MBP-HsC53$^{N\text{-}C(1\text{-}262, (KGSGSTSGSG),317\text{-}506)}$ | This study | | See Methods, Cloning procedures. Available on request to the corresponding authors. |
| Recombinant DNA reagent | MBP-AtC53$^{Y304A}$ | This study | | See Methods, Cloning procedures. Available on request to the corresponding authors. |
| Recombinant DNA reagent | MBP-AtC53$^{Y304A, 1A (W276A)}$ | This study | | See Methods, Cloning procedures. Available on request to the corresponding authors. |
| Recombinant DNA reagent | MBP-AtC53$^{Y304A, 2A (W287A)}$ | This study | | See Methods, Cloning procedures. Available on request to the corresponding authors. |
| Recombinant DNA reagent | MBP-AtC53$^{Y304A, 3A (W335A)}$ | This study | | See Methods, Cloning procedures. Available on request to the corresponding authors. |
| Recombinant DNA reagent | MBP-AtC53$^{Y304A, 12A (W276A, W287A)}$ | This study | | See Methods, Cloning procedures. Available on request to the corresponding authors. |
| Recombinant DNA reagent | MBP-AtC53$^{Y304A, 13A (W276A, W335A)}$ | This study | | See Methods, Cloning procedures. Available on request to the corresponding authors. |
| Recombinant DNA reagent | MBP-AtC53$^{Y304A, 23A (W287A, W335A)}$ | This study | | See Methods, Cloning procedures. Available on request to the corresponding authors. |
| Recombinant DNA reagent | MBP-AtC53$^{Y304A, 123A (W276A, W287A, W335A)}$ | This study | | See Methods, Cloning procedures. Available on request to the corresponding authors. |
| Recombinant DNA reagent | MBP-HsC53$^{1A(W269A)}$ | This study | | See Methods, Cloning procedures. Available on request to the corresponding authors. |
| Recombinant DNA reagent | MBP-HsC53$^{2A(W294A)}$ | This study | | See Methods, Cloning procedures. Available on request to the corresponding authors. |

*Appendix 1—key resources table continued*

| Reagent type (species) or resource | Designation | Source or reference | Identifiers | Additional information |
|---|---|---|---|---|
| Recombinant DNA reagent | MBP-HsC53$^{3A(W312A)}$ | This study | | See Methods, Cloning procedures. Available on request to the corresponding authors. |
| Recombinant DNA reagent | MBP-HsC53$^{12A(W269A, W294A)}$ | This study | | See Methods, Cloning procedures. Available on request to the corresponding authors. |
| Recombinant DNA reagent | MBP-HsC53$^{13A(W269A, W312A)}$ | This study | | See Methods, Cloning procedures. Available on request to the corresponding authors. |
| Recombinant DNA reagent | MBP-HsC53$^{23A(W294A, W312A)}$ | This study | | See Methods, Cloning procedures. Available on request to the corresponding authors. |
| Recombinant DNA reagent | MBP-HsC53$^{123A(W269A, W294A, W312A)}$ | This study | | See Methods, Cloning procedures. Available on request to the corresponding authors. |
| Recombinant DNA reagent | MBP-AtC53$^{IDR\ sAIM (Y304A, W276A, W287A, W335A)}$ | This study | | See Methods, Cloning procedures. Available on request to the corresponding authors. |
| Recombinant DNA reagent | MBP-HsC53$^{IDR\ sAIM (W269A, W294A, W312A)}$ | This study | | See Methods, Cloning procedures. Available on request to the corresponding authors. |
| Recombinant DNA reagent | MBP-HsUFL1 | This study | | See Methods, Cloning procedures. Available on request to the corresponding authors. |
| Recombinant DNA reagent | MBP-HsDDRGK1 | This study | | See Methods, Cloning procedures. Available on request to the corresponding authors. |
| Recombinant DNA reagent | GST-AtUFL1 | This study | | See Methods, Cloning procedures. Available on request to the corresponding authors. |
| Recombinant DNA reagent | GST-AtC53 | This study | | See Methods, Cloning procedures. Available on request to the corresponding authors. |
| Recombinant DNA reagent | MBP | This study | | See Methods, Cloning procedures. Available on request to the corresponding authors. |
| Recombinant DNA reagent | Ts-ATG8A | This study | | See Methods, Cloning procedures. Available on request to the corresponding authors. |
| Recombinant DNA reagent | Ts-AtDDRGK1$^{(24-298)}$ | This study | | See Methods, Cloning procedures. Available on request to the corresponding authors. |
| Recombinant DNA reagent | Ts-AtC53 | This study | | See Methods, Cloning procedures. Available on request to the corresponding authors. |
| Recombinant DNA reagent | HIS6-ATG8A | This study | | See Methods, Cloning procedures. Available on request to the corresponding authors. |
| Recombinant DNA reagent | HIS6-GABARAP | This study | | See Methods, Cloning procedures. Available on request to the corresponding authors. |
| Recombinant DNA reagent | HIS6-AtC53 | This study | | See Methods, Cloning procedures. Available on request to the corresponding authors. |

*Continued on next page*

*Appendix 1—key resources table continued*

| Reagent type (species) or resource | Designation | Source or reference | Identifiers | Additional information |
|---|---|---|---|---|
| Recombinant DNA reagent | HIS6-AtC53 $^{sAIM\ (Y304A,\ W276A,\ W287A,\ W335A)}$ | This study | | See Methods, Cloning procedures. Available on request to the corresponding authors. |
| Recombinant DNA reagent | HIS6-HsC53 | This study | | See Methods, Cloning procedures. Available on request to the corresponding authors. |
| Recombinant DNA reagent | HIS6-HsC53 $^{sAIM(W269A,\ W294A,\ W312A)}$ | This study | | See Methods, Cloning procedures. Available on request to the corresponding authors. |
| Recombinant DNA reagent | Strep-AtC53 | This study | | See Methods, Cloning procedures. Available on request to the corresponding authors. |
| Recombinant DNA reagent | Strep -AtC53$^{AIM\ (F48A,\ Y69A)}$ | This study | | See Methods, Cloning procedures. Available on request to the corresponding authors. |
| Recombinant DNA reagent | Strep -AtC53$^{AIM\ (Y69A,\ Y76A)}$ | This study | | See Methods, Cloning procedures. Available on request to the corresponding authors. |
| Recombinant DNA reagent | Strep -AtC53$^{AIM\ (F48A,\ Y69A,\ Y76A)}$ | This study | | See Methods, Cloning procedures. Available on request to the corresponding authors. |
| Recombinant DNA reagent | Strep -AtC53$^{AIM\ (W100A)}$ | This study | | See Methods, Cloning procedures. Available on request to the corresponding authors. |
| Recombinant DNA reagent | Strep -AtC53$^{AIM\ (Y304A)}$ | This study | | See Methods, Cloning procedures. Available on request to the corresponding authors. |
| Recombinant DNA reagent | Strep -AtC53$^{AIM\ (F48A,\ Y69A,\ Y76A,\ W100A,\ Y304A)}$ | This study | | See Methods, Cloning procedures. Available on request to the corresponding authors. |
| Recombinant DNA reagent | GST | This study | | See Methods, Cloning procedures. Available on request to the corresponding authors. |
| Transfected construct (*Arabidopsis thaliana*) | pUbi::C53-mCherry | this study | | See Methods, Cloning procedures. Available on request to the corresponding authors. |
| Transfected construct (*Arabidopsis thaliana*) | pUbi::C53-GFP | this study | | See Methods, Cloning procedures. Available on request to the corresponding authors. |
| Transfected construct (*Arabidopsis thaliana*) | pC53::C53-GFP | this study | | See Methods, Cloning procedures. Available on request to the corresponding authors. |
| Transfected construct (*Arabidopsis thaliana*) | pUbi::C53$^{sAIM(W276A,\ W287A,Y304A,W335A)}$ -GFP | this study | | See Methods, Cloning procedures. Available on request to the corresponding authors. |
| Transfected construct (*Arabidopsis thaliana*) | pUbi::UFL1-GFP | this study | | See Methods, Cloning procedures. Available on request to the corresponding authors. |

*Continued on next page*

*Appendix 1—key resources table continued*

| Reagent type (species) or resource | Designation | Source or reference | Identifiers | Additional information |
|---|---|---|---|---|
| Transfected construct (*Arabidopsis thaliana*) | pUbi::DDRGK1-GFP | this study | | See Methods, Cloning procedures. Available on request to the corresponding authors. |
| Transfected construct (*Arabidopsis thaliana*) | pUbi::C53-GFP | this study | | See Methods, Cloning procedures. Available on request to the corresponding authors. |
| Transfected construct (*Arabidopsis thaliana*) | pUbi::mCherry-ATG8A | this study | | See Methods, Cloning procedures. Available on request to the corresponding authors. |
| Transfected construct (*Arabidopsis thaliana*) | pUbi::GFP-ATG8A | this study | | See Methods, Cloning procedures. Available on request to the corresponding authors. |
| Transfected construct (*Arabidopsis thaliana*) | pUbi::GFP-ATG8B | this study | | See Methods, Cloning procedures. Available on request to the corresponding authors. |
| Transfected construct (*Arabidopsis thaliana*) | pUbi::GFP-ATG8C | this study | | See Methods, Cloning procedures. Available on request to the corresponding authors. |
| Transfected construct (*Arabidopsis thaliana*) | pUbi::GFP-ATG8D | this study | | See Methods, Cloning procedures. Available on request to the corresponding authors. |
| Transfected construct (*Arabidopsis thaliana*) | pUbi::GFP-ATG8E | this study | | See Methods, Cloning procedures. Available on request to the corresponding authors. |
| Transfected construct (*Arabidopsis thaliana*) | pUbi::GFP-ATG8F | this study | | See Methods, Cloning procedures. Available on request to the corresponding authors. |
| Transfected construct (*Arabidopsis thaliana*) | pUbi::GFP-ATG8G | this study | | See Methods, Cloning procedures. Available on request to the corresponding authors. |
| Transfected construct (*Arabidopsis thaliana*) | pUbi::GFP-ATG8H | this study | | See Methods, Cloning procedures. Available on request to the corresponding authors. |
| Transfected construct (*Arabidopsis thaliana*) | pUbi::GFP-ATG8I | this study | | See Methods, Cloning procedures. Available on request to the corresponding authors. |
| Transfected construct (*Arabidopsis thaliana*) | pUbi::IRE1B-YFP x pRPS5a::C53-tagRFP | this study | | See Methods, Cloning procedures. Available on request to the corresponding authors. |
| Transfected construct (*Homo sapiens*) | psPAX2 | Addgene | 12260 | Didier Trono |

*Continued on next page*

*Appendix 1—key resources table continued*

| Reagent type (species) or resource | Designation | Source or reference | Identifiers | Additional information |
|---|---|---|---|---|
| Transfected construct (*Homo sapiens*) | pMD2.G | Addgene | 12259 | Didier Trono |
| Transfected construct (*Homo sapiens*) | C53 shRNA in pLKO1 | Honglin Li Wu et al. Cell Res (2013). | | |
| Transfected construct (*Homo sapiens*) | peGFP(N2)-HsC53-GFP | This study | | See Methods, Cloning procedures. Available on request to the corresponding authors. |
| Transfected construct (*Homo sapiens*) | peGFP(N2)-AtC53-GFP | This study | | See Methods, Cloning procedures. Available on request to the corresponding authors. |
| Transfected construct (*Homo sapiens*) | peGFP(N2)-HsC53$^{sAIM}$-GFP | This study | | See Methods, Cloning procedures. Available on request to the corresponding authors. |
| Transfected construct (*Homo sapiens*) | peGFP(N2)-AtC53 | This study | | See Methods, Cloning procedures. Available on request to the corresponding authors. |
| Transfected construct (*Homo sapiens*) | pmCherry(N2)-HsC53-mCherry | This study | | See Methods, Cloning procedures. Available on request to the corresponding authors. |
| Transfected construct (*Homo sapiens*) | pmCherry-GABARAP-mCherry | Fumiyo Ikeda | | |
| Transfected construct (*Homo sapiens*) | mRFP-LAMTOR1 | Sascha Martens | | |
| Transfected construct (*Homo sapiens*) | ER-K20 | Addgene Wang et al. Cell Res. (2020) | 133861 | |
| Transfected construct (*Homo sapiens*) | ERAD-C (pEGFP-GFP: CFTR$_{\Delta F508}$ ) | Ron R. Kopito Leto et al. Mol. Cell (2019) | | |
| Transfected construct (*Homo sapiens*) | ERAD-L (pcDNA3-NHK-GFP) | Ron R. Kopito Leto et al. Mol. Cell (2019) | | |
| Transfected construct (*Homo sapiens*) | ERAD-M (pMCB497-pTRE-INSIG1-GFP) | Ron R. Kopito Leto et al. Mol. Cell (2019) | | |
| Transfected construct (*Homo sapiens*) | pcDNA3-Erdj3-GFP-3Gly | Maya Schuldiner *Ast et al., 2016* | | |

*Continued on next page*

*Appendix 1—key resources table continued*

| Reagent type (species) or resource | Designation | Source or reference | Identifiers | Additional information |
|---|---|---|---|---|
| Chemical compound, drug | Tunicamycin | SCBT | sc-3506 | |
| Chemical compound, drug | Torin | SCBT | sc-396760 | |
| Chemical compound, drug | Bafilomycin A1 | Abcam | ab120497 | |
| Chemical compound, drug | 4µ8C | Sigma Aldrich | SML0949 | |
| Chemical compound, drug | KIRA6 | MedChemExpress | HY-19708 | |
| Chemical compound, drug | Anisomycin (ANS) | Sigma Aldrich | A5862-0.5ml | |
| Chemical compound, drug | DTT | Sigma Aldrich | 43815 | |
| Chemical compound, drug | Concanamycin-A (conA) | Santa Cruz | sc-202111A | |
| Chemical compound, drug | Cyclopiazonic acid (CPA) | Santa Cruz | sc-201510 | |
| Chemical compound, drug | Kifunensine (kif) | Santa Cruz | sc-201364A | |
| Chemical compound, drug | Thapsigargin (Tg) | Santa Cruz | sc-24017 | |
| Chemical compound, drug | CB-5083 | Selleckchem | # S8101 | |
| Chemical compound, drug | Harringtonine | Santa Cruz | sc-204771 | |
| Chemical compound, drug | Anisomycin | Sigma Aldrich | 176880–10 MG | |
| Chemical compound, drug | Puromycin | Sigma Aldrich | P8833-10MG | |
| Chemical compound, drug | Emetine | Sigma Aldrich | E2375-250MG | |
| Strain, strain background (*E. coli*) | DH5α | In-house facility | | Vienna BioCenter |
| Strain, strain background (*E. coli*) | BL21 (DE3) | In-house facility | | Vienna BioCenter |

*Continued on next page*

*Appendix 1—key resources table continued*

| Reagent type (species) or resource | Designation | Source or reference | Identifiers | Additional information |
|---|---|---|---|---|
| Strain, strain background (*E. coli*) | Rosetta2 (DE3) pLysS | In-house facility | | Vienna BioCenter |
| Strain, strain background (*E. coli*) | GV3101 (pSoup) | In-house facility | | Vienna BioCenter |
| Software, algorithm | CLC main work bench 7 | Qiagen | | Cloning |
| Software, algorithm | Zen Software | Carl Zeiss | | Microscopy |
| Software, algorithm | Image J (Fiji) | NIH | | Image Quantification |
| Software, algorithm | Prism 8 | Graph Pad | | Statistics |
| Software, algorithm | Image Lab | BioRad | | Western Blot Analysis |
| Software, algorithm | Adobe Illustrator 2020 | Adobe Inc | | Graphics editing |
| Software, algorithm | RStudio 1.2.5019 | RStudio, Inc | | Graph plotting |
| Other | GFP-Trap | Chromotek | Gta-20 | |
| Other | RFP-Trap | Chromotek | Rta-20 | |
| Other | Glutathion Sepharose 4 | GE Healthcare | 17-5132-01 | |
| Other | Pierce Glutathione Magnetic Agarose Beads | Thermo Scientific | 78601 | |
| Other | HisTrap FF 5 ml | GE Healthcare | 17525501 | |
| Other | HisTrap FF 1 ml | GE Healthcare | 17531901 | |
| Other | Resource Q 6 ml | GE Healthcare | 17117901 | |
| Other | Resource S 6 ml | GE Healthcare | 17118001 | |
| Other | HiPrep 26/10 Desalting | GE Healthcare | 17508701 | |
| Other | HiLoad 16/600 Superdex 75 pg | GE Healthcare | 28989333 | |
| Other | HiLoad 16/600 Superdex 200 pg | GE Healthcare | 28989335 | |
| Other | GSTrap FF | GE Healthcare | 17513101 | |
| Other | Streptavidin-HRP Conjugate | GE Healthcare | GERPN1231-100UL | 1:1000 |

