## [Decision Letter]

**Acceptance summary:**

While having been intensely studied in yeast or mammalian cells, autophagy receptors remain less well understood in plants. In this study, the authors identify C53 as a conserved autophagy receptor from plants to human cells. C53 recognizes a shuffled ATG8-interaction motif to bind ATG8. While C53 does not play a role in general autophagy, it is recruited to clogged ER translocons during times of ER stress. C53 accumulates at the ER via DDRGK1 and UFL1, thus coordinating UFM1-modification of RPS26 in stalled ribosomes with formation of an autophagophore. Consistent with these observations, mutants in C53 or the plant UFMylation machinery were specifically sensitive to ER stress. This paper therefore provides important insight into conserved mechanisms underlying selective ER-phagy from plants to humans.

**Decision letter after peer review:**

Thank you for submitting your article "A cross-kingdom conserved ER-phagy receptor maintains endoplasmic reticulum homeostasis during stress" for consideration by *eLife*. Your article has been reviewed by three peer reviewers, one of whom is a member of our Board of Reviewing Editors, and the evaluation has been overseen by Suzanne Pfeffer as the Senior Editor. The reviewers have opted to remain anonymous.

The reviewers have discussed the reviews with one another and the Reviewing Editor has drafted this decision to help you prepare a revised submission.

Summary:

The endoplasmic reticulum (ER) orchestrates the folding and secretion of a large number of proteins. Protein misfolding in the ER can be corrected by the ubiquitin-proteasome system, as well as by autophagy of parts of the ER, a process referred to as ER-phagy. While several adaptors that link ER to the autophagy machinery have been described, how this process is being regulated is only beginning to be understood.

In this manuscript, Stephani and co-workers identify a protein, C53, that is recruited to ATG8 proteins upon ER stress. C53 recognizes ATG8, a hallmark modification of autophagy, through a novel "shuffled ATG8-interacting motif". C53 localization changes in cells expressing a reporter for stalled ribosomes, and cells lacking C53 are particularly sensitive to ER-stress that is elicited by aberrant co-translational protein synthesis and ER targeting. The experiments describing the interaction between C53 and ATG8, both in plants and human cells, have been performed well and establish C53 as a bona fide ATG8-binding protein. Also, the change in localization of C53 upon specific ER stresses is very interesting and indicative of a potential function of this protein in ER-phagy. However, there are several major issues that would need to be addressed before this manuscript could be accepted for publication in *eLife*. We recognize that these issues require significant experimentation, yet due to Covid-19, we will not impose a time limit to the authors on when to resubmit to *eLife*.

Essential, major issues that need to be addressed with experiments prior to publication are:

1) The authors do not show that C53 localizes to ER upon stress, a prerequisite for a specific ER-phagy receptor. They also do not demonstrate that C53 is required for specific ER-phagy. These questions would need to be addressed through better localization analyses (i.e. co-localization of C53 and ER markers) and flux studies (to show that bulk ER proteins are turned over in a C53- and autophagy-dependent manner). Moreover, the authors should include a more careful investigation of the role of the UPR in these processes.

2) The authors would need to improve their investigation of C53's role in turning over stalled ribosomes at the ER. The current data is not sufficient and would need to be supported by flux assays that show reporter turnover in a C53- and autophagy-dependent manner in a pulse-chase setting.

3) Given their focus and previous publications, the authors would also need to show that the UFM1 machinery is required for C53's role in ER-phagy. They also should perform sequential IPs to show that C53 is really part of a ternary complex with UFL1 and DDRGK1, as explicitly stated by their model.

4) In addition to these experimental revisions, this manuscript requires substantial editing to allow readers understand the experiments and their significance. The reasoning between experiments needs to be explained in more detail. For example, the authors need to explicitly discuss concentrations (i.e. are protein levels used for competition experiments comparable to cellular concentrations?); moreover, they should describe how many AIMs are outside of the IDR region, how were they identified, and how many were tested; finally, the authors should describe better what they mean by a "shuffled AIM" – is this a new ATG8-interaction motif or do they believe it is just a minor variation of the standard recognition element? These are examples – the whole manuscript would greatly benefit from more explicit explanations of experiments to make it more accessible to a larger audience.

---

## [Author Response]

Essential, major issues that need to be addressed with experiments prior to publication are:1) The authors do not show that C53 localizes to ER upon stress, a prerequisite for a specific ER-phagy receptor.

In the previous version of the manuscript, we had already analyzed the localization of C53 in plants by immunogold labeling with two different antibodies and live cell imaging using confocal microscopy under normal and stress conditions.

Our electron micrographs obtained from non-stressed cells already suggested C53 is in close proximity to the ER (original Figure 2—figure supplement 3). We have now performed a more detailed electron microscopy analysis demonstrating that C53 localizes to the ER and delivered to the vacuole, but does not localize to other subcellular compartments such as multivesicular bodies, Golgi apparatus or mitochondria. (updated Figure 2—figure supplement 3). In addition, we carried out colocalization assays with (i) a canonical ER marker (HDEL-GFP) (now presented in Figure 2—figure supplement 1A), (ii) DDRGK1, a transmembrane domain containing protein that localize uniformly at the ER (Liang et al., 2020) (original Figure 8 and new Figure 8—figure supplement 2), and (iii) UFL1, the ufmylation E3 ligase residing in the ER (original Figure 8). This confocal microscopy analysis also shows that upon ER stress, C53 specifically co-localizes with both DDRGK1 and UFL1 (presented in Figure 8C and the new Figure 8—figure supplement 2).

In addition, we have already showed in the previous version of the manuscript that C53 puncta co-localize with the ER targeted ribosome stalling substrate (presented in Figure 7—figure supplement 2) in HeLa cells. This substrate has been shown to be trapped in Sec61 translocons that are on the ER (Wang et al., 2020).

Altogether the new additional electron microscopy and confocal microscopy data and the previously presented imaging data suggest C53 is recruited to ER during stress.

They also do not demonstrate that C53 is required for specific ER-phagy. These questions would need to be addressed through better localization analyses (i.e. co-localization of C53 and ER markers) and flux studies (to show that bulk ER proteins are turned over in a C53- and autophagy-dependent manner).

To demonstrate that C53 is required for ER-phagy, we performed the suggested confocal microscopy and autophagic flux assays. Using confocal microscopy and western blot-based flux assays, we show that ER resident protein DDRGK1, but not the abundant ER resident proteins SMT1, BIPs or Calnexins are degraded via C53-mediated autophagy (Presented in Figure 8—figure supplement 2). In addition, DDRGK1 form punctate structures that colocalize with ATG8A in a C53 dependent manner (Presented in Figure 8—figure supplement 2). We also have presented data in the previous version of the manuscript that showed that C53 mediates lysosomal delivery of the ER targeted ribosome stalling construct in HeLa cells (Figure 7—figure supplement 2). Together, our data confirm that C53 mediates ER-phagy, however functions differently than the known ER-phagy receptors. Rather than shaping the ER upon starvation or ER-stress, it is specifically involved in recycling stalled nascent chains at the ER. To clarify this point, we added a paragraph in the Discussion, comparing C53 with other ER-phagy receptors.

With regards to its mechanism, our data suggest that C53 is not involved in recycling of ER tubules or sheets per se but rather enables a quality control pathway to recycle arrested polypeptides of ER-stalled ribosomes. Moreover, the quality-control activity of C53 seems to be highly specific: even though both the Clogger and ER-K20 constructs are localized at Sec61 translocons, only ER-K20, a co-translational ribosome stalling construct is capable to activate C53 and get degraded by C53 autophagy.

To test specific induction of C53 upon ribosome stalling, we tested its activation by different translation inhibitors. Our new data show that C53 is activated by various translation elongation inhibitors, but not by drugs that inhibit initiation of translation. This phenotype contrasts to the occurrence of ATG8 puncta, which are activated with all translation inhibitors (new Figure 7—figure supplement 2). Together, these data indicate that C53 is not involved in reticulophagy or recycling of ER sheets, but in recycling stalled translation products. Consistently, quantitative proteomics data presented in the original manuscript (Figure 6) revealed that proteins accumulating in C53 deficient cells are mostly proteins that are synthesized in the ER-bound ribosomes that could trigger ribosome stalling during translational elongation.

Moreover, the authors should include a more careful investigation of the role of the UPR in these processes.

Plants utilize two major UPR branches, the Ire1 and bZIP17/28 (homologous to ATF6), however lack PERK homologs (Pastor-Cantizano, et al., 2020). To address the connection with the plant UPR system, we thus focused our analysis on the Ire1 and bZIP17/28 ER stress sensors. In the original version of the manuscript, we already presented data showing that C53 protein levels are elevated in bZIP17/28 mutants and that C53 autophagic flux is not affected in neither ire1a/b double mutant or bzip17/28 double mutants. Furthermore, we presented data showing that C53 still forms puncta in response to ER stress. We also corroborated these findings using Ire1 inhibitors in HeLa cells. To further confirm these data, we carried out further experiments showing that C53 puncta are still induced upon ER stress in bZIP28/bZIP60 mutants. Moreover, C53 puncta do not colocalize with IRE1 oligomers that form upon ER stress (updated Figure 7). As C53 is highly conserved and plants only employ two UPR pathways, our data conclusively show that C53 mediated autophagy is not activated by UPR sensors.

We also would like to note that all plant experiments are performed with stable transgenic lines. As there has been several examples, where cell culture studies were not transferrable to organisms, the use of stable transgenic organisms – as in the present study – is highly beneficial for obtaining robust and reproducible results.

2) The authors would need to improve their investigation of C53's role in turning over stalled ribosomes at the ER. The current data is not sufficient and would need to be supported by flux assays that show reporter turnover in a C53- and autophagy-dependent manner in a pulse-chase setting.

As described in the original manuscript, our quantitative proteomics data suggest that C53 does not degrade stalled ribosomes. New data presented in Figure 8—figure supplement 2B confirm our notion. In contrast to DDRGK1, neither 40S nor 60S ribosomal subunits are degraded via C53 (new Figure 8—figure supplement 2). Moreover, we show that abundant ER proteins, such as calnexin or the BIP chaperones, are also not degraded via the C53 pathway. However, a comprehensive analysis of cargo selection and recruitment within the ER is beyond the scope of this manuscript.

3) Given their focus and previous publications, the authors would also need to show that the UFM1 machinery is required for C53's role in ER-phagy.

In the original version of our manuscript, we have presented data illustrating that upon ER stress, UFL1 forms puncta that colocalize with C53 and that these puncta are delivered to the vacuole. We have also shown that in stable ufl1 mutant plants, C53 puncta do not form upon ER stress (Figure 8—figure supplement 1). We also presented data showing that all ufmylation pathway mutants are sensitive to ER stress but not starvation (Figure 10—figure supplement 1). To provide further experimental evidence for the proposed mechanism, we carried out an autophagic flux experiment showing that C53 flux is affected in all ufmylation mutants (Figure 8—figure supplement 1F). Together, our data confirm the connection of UFM1 machinery and C53-mediated ER-phagy.

They also should perform sequential IPs to show that C53 is really part of a ternary complex with UFL1 and DDRGK1, as explicitly stated by their model.

We based our conclusion that C53-UFL1-DDRGK1 forms a tripartite receptor complex (now modified to heteromeric receptor complex) based on in vitro pull downs, in vivo pull downs, yeast two hybrid assays and colocalization studies. To further support our model, we have performed additional pull-down experiments, where we were able to pull down both DDRGK1 and C53 in a single UFL1 pull down (Figure 8—figure supplement 1A). Additionally, we have included DDRGK1, UFL1 and C53 affinity purification mass spectrometry results under normal and stress conditions that showed that all three proteins associate with each other, and this interaction becomes stronger upon ER stress treatments (Supplementary file 5).

4) In addition to these experimental revisions, this manuscript requires substantial editing to allow readers understand the experiments and their significance. The reasoning between experiments needs to be explained in more detail. For example, the authors need to explicitly discuss concentrations (i.e. are protein levels used for competition experiments comparable to cellular concentrations?).

We have now significantly edited the manuscript to explain the experiments more clearly. We have extended the text to better explain the rationale behind the experimental design and details, and our conclusions. We have also extended the Discussion section to compare and contrast C53 mediated autophagy to other known ER-phagy receptors. We believe the revised text allows readers to follow and judge our findings much better.

Regarding the protein levels, for the in vivo competition experiments, we used native antibodies against ATG8 and UFM1. For C53 flux analyses, we performed assays using native C53 antibodies, native promoter driven C53 transgenic lines and Ubiquitin10 promoter driven transgenic lines. In all cases, we have obtained comparable results. Nevertheless, we have added new confocal microscopy data that shows that “native” promoter driven C53 forms similar levels of puncta when compared to “Ubiquitin10” promoter driven C53 lines (Presented in Figure 2—figure supplement 1).

Moreover, they should describe how many AIMs are outside of the IDR region, how were they identified, and how many were tested.

We are absolutely confident of our assessment that C53 interacts with ATG8 via the sAIMs. We have reached to this conclusion by (i) truncation experiments and ITC assays that showed that the IDR of both human and Arabidopsis C53 is necessary and sufficient to bind to ATG8 (Figure 4), (ii) native mass spectrometry experiments that showed that C53^sAIM^ mutants are unable to form a complex with ATG8 (now Figure 4—figure supplement 2), (iii) SPR experiments with full length C53 proteins that showed that C53^sAIM^ mutants are unable to bind ATG8 (Figure 5), and (iv) complementation experiments in Arabidopsis and HeLa cells that showed that C53^sAIM^ mutants are unable to form puncta that co-localize with ATG8 (Figure 5). CD spectroscopy and western blot experiments showed that sAIM mutants fold similar to the wild type proteins and are equally stable. So, the lack of puncta is not due to protein instability but loss of ATG8 interaction. Nevertheless, to address the raised point, we mutagenized all potential canonical AIM sites in the Arabidopsis C53 protein one by one and in a combinatorial manner. These experiments confirm that the canonical AIMs of C53 do not mediate ATG8 binding (Presented in Figure 4—figure supplement 1).

Finally, the authors should describe better what they mean by a "shuffled AIM" – is this a new ATG8-interaction motif or do they believe it is just a minor variation of the standard recognition element? These are examples – the whole manuscript would greatly benefit from more explicit explanations of experiments to make it more accessible to a larger audience.

As clearly indicated in the original manuscript, our in vitro data suggest sAIM binds ATG8 via the canonical LIR-docking site. However, while the critical AIM residues (IDWG) are maintained, their order is mixed up relative to the standard motif (should be W-D,G-I) and could not be reverted. Therefore, we decided to use the term “shuffled AIM” rather than defining it as a new ATG8 interacting motif. We modified the text to clarify this.